# The analogue method for precipitation prediction: finding better analogue situations at a sub-daily time step

Pascal Horton[1,2], Charles Obled[3], and Michel Jaboyedoff[1]

[1]University of Lausanne, Institute of Earth Sciences, Lausanne, Switzerland
[2]University of Bern, Oeschger Centre for Climate Change Research, Institute of Geography, Bern, Switzerland
[3]Université de Grenoble-Alpes, LTHE, Grenoble, France

*Correspondence to:* Pascal Horton (pascal.horton@alumnil.unil.ch)

**Abstract.** Analogue methods (AMs) predict local weather variables (predictands), such as precipitation, by means of a statistical relationship with predictors at a synoptic scale. The analogy is generally assessed on gradients of geopotential heights first, in order to sample days with a similar atmospheric circulation. Other predictors, such as moisture variables, can also be added in a successive level of analogy.

5      The search for candidate situations for a given target day is usually undertaken by comparing the state of the atmosphere at fixed hours of the day for both the target day and the candidate analogues. The main reason is the use of daily precipitation time series due to the length of their available archives, and the unavailability of equivalent archives at a finer time step. However, it is unlikely for the best analogy to occur at the very same hour, while it may be found with a time shift of some hours as it can occur at a different time of day. In order to assess the potential for finding better analogues at a different hour, a moving time 10  window (MTW) has been introduced.

     The MTW resulted in a better analogy in terms of the atmospheric circulation, with improved values of the analogy criterion on the entire distribution of analogue dates. The improvement was found to grow with the analogue rank due to an accumulation of better analogues in the selection. A seasonal effect has also been identified, with larger improvements in winter than in summer, supposedly due to stronger diurnal cycles in summer that favour predictors at the same hour for the target and analogue 15  days.

     The impact of the MTW on prediction skill has been assessed by means of a sub-daily precipitation series transformed into moving 24h totals at 6h time steps. The prediction skill was found to have improved by the MTW, and even to a greater extent after recalibrating the AM parameters. Moreover, the improvement was greater for days with heavy precipitation, which are generally related to more dynamic atmospheric situations where timing is more specific and which are fewer in the meteoro-20  logical archive.

     However, in order to produce quantitative precipitation predictions the MTW requires sub-daily precipitation time series, which are generally available for a shorter period than daily precipitation archives. Therefore, two simple temporal disaggregation methods were assessed in order to make longer archives usable with the MTW. The assessed approaches were not successful, emphasizing the need to use time series with an appropriate chronology. These should be available in the near

future, either by means of growing archives of measurements or by the establishment of regional precipitation reanalysis data at sub-daily time step.

The use of the MTW in the AM can already be considered now for several applications in different contexts, may it be for operational forecasting or climate-related studies.

## 5  1  Introduction

Analogue methods (AMs) are based on the hypothesis that two relatively similar synoptic situations may produce similar local effects (Lorenz, 1956, 1969). They are statistical downscaling methods (Maraun et al., 2010) and consist of finding past situations that are similar to the target day of interest in terms of the atmospheric circulation or other synoptic predictors. The local weather variables of interest (predictand) that were observed at the analogue dates are used to construct a probabilistic 10 prediction for the target day (Duband, 1970; Zorita and Storch, 1999). Multiple variations of the method exist (a non-exhaustive listing can be found in Ben Daoud et al., 2016). The AMs are used for operational precipitation forecasting, either in the context of weather forecasting, flood forecasting, or hydropower production (e.g., Guilbaud, 1997; Bontron and Obled, 2005; Hamill and Whitaker, 2006; Desaint et al., 2008; García Hernández et al., 2009; Bliefernicht, 2010; Marty, 2010; Marty et al., 2012; Horton et al., 2012; Obled, 2014; Hamill et al., 2015; Ben Daoud et al., 2016), as well as for precipitation downscaling from 15 a climate perspective (e.g. Radanovics et al., 2013; Chardon et al., 2014; Dayon et al., 2015). Other applications focus on temperatures (Radinovic, 1975; Woodcock, 1980; Kruizinga and Murphy, 1983; Delle Monache et al., 2013; Caillouet et al., 2016), wind (Gordon, 1987; Delle Monache et al., 2013, 2011; Vanvyve et al., 2015; Alessandrini et al., 2015b; Junk et al., 2015b, a), solar power (Alessandrini et al., 2015a; Bessa et al., 2015), snow avalanches (Obled and Good, 1980; Bolognesi, 1993), insolation (Bois et al., 1981), and the trajectories of tropical cyclones (Keenan and Woodcock, 1981; Sievers et al., 20 2000; Fraedrich et al., 2003).

The spatial transferability of the method is analysed in Chardon et al. (2014) and Radanovics et al. (2013). A great advantage of AMs is that they create realistic precipitation patterns, since they are based on observed situations with consistent spatial distribution, as long as the analogue dates chosen for a region are the same. Moreover, they can provide multivariate predictions that are physically consistent (Raynaud et al., 2016). Their temporal transferability has also been the topic of recent studies for 25 past or future climates (Dayon et al., 2015; Caillouet et al., 2016).

The method requires two different archives. The first is a meteorological archive describing the state of the atmosphere at a synoptic scale, such as reanalysis datasets. The second is an archive of the target variable to be forecast, here precipitation. This archive is made of precipitation cumulated over a certain time duration, most often daily but sometimes sub-daily, either at a target station or integrated over a target catchment. Obviously, the period to be used is limited to the smallest period common 30 to the two archives.

Ruosteenoja (1988) and Van Den Dool (1994) have analysed the influence of the length of the meteorological archive on the quality of the analogy. They highlighted a three-way relationship between the quality of the analogy, the archive length, and the size of the spatial domain (or degrees of freedom): errors increase with a bigger domain, but decrease with a longer

archive. They demonstrated that it is not possible to find good analogues over the whole northern hemisphere with a 100 year archive (and even for much higher orders of magnitude). Hopefully, it appears that for the aim of predicting precipitation over point stations or catchments of some 100 or 1000 km², there is no need to consider meteorological domains larger than 10° to 20°. For that reason, smaller spatial windows are always considered when searching for analogues, and the archive length is maximized.

Therefore, due to the availability of long precipitation archives at a daily time step that have no equivalent at a finer resolution, AMs are usually implemented on a daily basis. Consequently, the analogue situations are assessed by comparing predictors at fixed hours of the day. However, it can be expected that the analogy of the synoptic situations does not occur systematically at the same time of the day and that better candidates can be found by shifting to a different hour. With this assumption, a moving time window (MTW) was introduced to allow the search for candidates at different hours of the day. Previous tests showed the benefit, in terms of analogy criterion values, of searching for analogue synoptic situations at a finer time step, but without assessing the impact on the prediction skill (Finet et al., 2008).

The MTW finds analogue situations at different hours of the day, which can also be seen as an inflation of the archive length. However, despite having $x$ time more candidate situations, the quantity of new information is not expected to be as important as a $x$ time longer archive due to significant correlation between successive situations within the same day. Nevertheless, if the MTW can improve the prediction skill of the AM, it means that it does extract new information from the archive. Therefore, if the reduction of the archive length needed by the MTW, due to the reduced availability of a sub-daily precipitation time series, is expected to decrease the AM performance, the archive inflation brought by the MTW is expected to contribute to an increase in performance.

Other possibilities exist for increasing the prediction skill of the AMs. A classical approach is to add new predictors or new successive levels of analogy (see e.g. Horton, 2012; Ben Daoud et al., 2016; Caillouet et al., 2016). AMs can also be combined with other methods (see e.g. Chardon et al., 2014). Another possibility is to use a global optimization technique, such as genetic algorithms, in order to better optimize the method and to add new parameters (Horton et al., 2016). However, the MTW technique is not in competition with other approaches and can be combined with these. Indeed, as most of them rely on the atmospheric circulation in the first level of analogy, the application of a MTW should lead to similar conclusions.

Section 2 presents the context of the study as well as the data and methods, including the proposed MTW technique. The impact of the reduction of the archive and the improvements brought by the MTW are assessed in Sect. 3. The benefits of introducing a MTW were assessed first in regards to the analogy criterion improvement between synoptic situations (Sect. 3.2) and then in terms of precipitation prediction skill (Sect. 3.3). Finally, the results are discussed in Sect. 4, and the conclusions are found in Sect. 5.

## 2 Data and methods

### 2.1 Study area and data

The study area is the upper Rhône catchment in Switzerland. Precipitation time series come from six automatic weather stations, viz., Ulrichen, Zermatt, Visp, Montana, Sion, and Aigle (Fig. 1) that are subject to various meteorological influences (Horton et al., 2012). The data were available at an hourly time step for 25 years (1982–2007) and were also obtained at a daily time step (from 6:00 UTC to 6:00 UTC the next day) for 47 years (1961–2008). Due to the low density of weather stations with high temporal resolution and long archives, no spatially aggregated rainfall was processed. The results will hereafter be presented arbitrarily for the Ulrichen station but are equivalent for all stations.

Synoptic-scale variables, used as predictors, were extracted from the NCEP/NCAR reanalysis 1 dataset (Kalnay et al., 1996) with a 6h temporal resolution, 17 pressure levels, and a spatial resolution of 2.5°. This dataset is now relatively old, but it is not expected to affect the conclusions of the present study (see discussion in Sect. 4.6).

### 2.2 The considered analogue method

The first considered AM is based on the analogy of the atmospheric circulation only (Table 1, Obled et al., 2002; Bontron and Obled, 2005). Searching for analogue situations to a target day starts by a seasonal stratification through a preselection step of the potential candidates for analogy. The restriction is a search for analogue days within a 4-month window centred on the target date for every year of the archive. The similarity of the atmospheric circulation of the target date with every day of the preselection is assessed by processing the Teweles and Wobus (1954) score (S1) that is a comparison of gradients on geopotential heights over a certain spatial window and at certain hours:

$$S_1 = 100 \frac{\sum_{i}^{m} |\Delta \hat{z}_i - \Delta z_i|}{\sum_{i}^{m} max(|\Delta \hat{z}_i|, |\Delta z_i|)} \tag{1}$$

where $\Delta \hat{z}_i$ is the forecast geopotential height difference between the $i$th pair of adjacent points from the grid of the target situation, $\Delta z_i$ is the corresponding observed geopotential height difference in the candidate situation, and $m$ is the number of pairs of adjacent points in the grid. The differences are processed separately in both directions. With smaller S1 values, there is greater similarity in the pressure fields.

The predictor variables extracted from reanalysis datasets are considered at different hours of the day. Based on Bontron and Obled (2005), geopotential heights are compared at 1000 hPa (Z1000) at 12:00 UTC and 500 hPa (Z500) at 24:00 UTC. The time of the day at which the predictors are selected is found by Bontron (2004) to have a significant influence.

Then, $N_1$ dates with the lowest values of S1 are considered as analogues to the target day, $N_1$ being a parameter to calibrate (see Sect. 2.6). Finally, the daily observed precipitation amount of the corresponding dates provide the empirical conditional distribution considered as the probabilistic prediction for the target day. This method will be named 2Z.

The second reference method (2Z-2MI, Table 2) adds a subsequent level of analogy with moisture variables, compared by means of the root-mean-square error (RMSE):

$$E_{RMS} = \sqrt{\frac{1}{n}\sum_{i=1}^{n}(\hat{v}_i - v_i)^2} \tag{2}$$

where $\hat{v}_i$ is the $i$th predictor value from the grid of the target situation, $v_i$ is the corresponding observed value in the candidate situation, and $n$ is the number of points in the grid.

The additional predictor is a moisture index composed of the product of the total precipitable water (TPW) with the relative humidity at 850 hPa (RH850) (Bontron, 2004). When adding a second level of analogy, $N_2$ dates are subsampled from the $N_1$ analogues on the atmospheric circulation, resulting in a smaller number of analogue situations. When a second level of analogy is added, a higher number of $N_1$ analogues is kept on the first level.

More complex AMs exist with additional predictors (see e.g. Horton, 2012; Ben Daoud et al., 2016; Caillouet et al., 2016). The MTW can also be applied to these, as they generally rely on a similarity of the atmospheric circulation in the first level of analogy. However, it is easier to interpret the impact of the MTW using more basic methods.

## 2.3 Performance score

In order to assess the performance of AMs, the continuous ranked probability score (CRPS, Brown, 1974; Matheson and Winkler, 1976; Hersbach, 2000) is often employed (see, e.g., Bontron, 2004; Bontron and Obled, 2005; Ben Daoud et al., 2008; Horton et al., 2012; Marty et al., 2012; Radanovics et al., 2013; Chardon et al., 2014; Junk et al., 2015b; Ben Daoud et al., 2016; Caillouet et al., 2016). It allows the evaluation of the predicted cumulative distribution functions $F(y)$ of the precipitation values $y$ from analogue situations compared to the observed value $y^0$. A better prediction has a smaller score. It is defined as follows:

$$S_{\text{CRP}} = \int_{-\infty}^{+\infty} \left[ F(y) - \text{H}(y - y^0) \right]^2 \text{d}y, \tag{3}$$

where $\text{H}(y - y^0)$ is the Heaviside function that is null when $y - y^0 < 0$ and has the value of 1 otherwise.

In order to compare the value of the score in regard to a reference, one often considers its skill score expression and uses the climatological distribution (i.e., the distribution of all precipitation values from the corresponding archive) as the reference. The continuous ranked probability skill score (CRPSS) is defined as follows:

$$S_{\text{SCRP}} = \frac{S_{\text{CRP}} - S_{\text{CRP}}^r}{S_{\text{CRP}}^p - S_{\text{CRP}}^r} = 1 - \frac{S_{\text{CRP}}}{S_{\text{CRP}}^r} \tag{4}$$

where $S_{\text{CRP}}^r$ is the $S_{\text{CRP}}$ value for the reference (climatological distribution) and $S_{\text{CRP}}^p$ is for a perfect prediction (which implies $S_{\text{CRP}}^p = 0$). A better prediction is characterized by an increase in CRPSS: $S_{\text{SCRP}} = 1$ for a perfect prediction and $S_{\text{SCRP}} < 0$ for a prediction with a lower skill than the reference.

## 2.4 The moving time window (MTW) approach

The moving time window (MTW) technique aims at finding better analogue situations at different hours of the day rather than comparing the predictors at the same fixed hours. If the target situation is kept at 12:00 and 24:00 UTC for Z1000 and Z500 respectively, candidate situations are not only considered at 12:00 and 24:00 UTC, but at other hours by allowing a time shift. Therefore, instead of looking for analogues at a 24h time step, they are sought at the time step matching the predictor temporal resolution which is a 6h time step in this study (Fig. 2).

The target situations and their corresponding observed precipitation values do not change because the prediction is still established on a daily basis for a fixed period of the target day (6:00 UTC to 6:00 UTC the next day) as before. The difference is that the candidates are 4 times as many (even though they are not fully independent) as in the conventional approach. No constraint was added in order to restrict the selection of multiple analogues within the same candidate dates.

In order to assess the benefit of searching for analogue situations at a sub-daily time step for quantitative precipitation
prediction, an appropriate precipitation series is required. On the basis of high temporal resolution time series (Sect. 2.1), 24h totals were processed at a 6h time step by means of a moving 24h total. Since sub-daily precipitation time series are available on a shorter period than daily ones, the loss of performance resulting from a reduction of a 47 year archive (1961–2008) to 25 years (1982–2007) can be expected. However, the competition between the loss of performance due to a smaller archive length and the potentially better analogy between target and MTW candidates needs to be assessed.

## 2.5 Reconstruction of a longer precipitation archive

It would be profitable to be able to apply the MTW to a longer archive (here, 1961–2008), rather than being limited to the reduced period where the sub-daily precipitation data are available (cf. previous section). Therefore, in order to reconstruct a longer archive of moving 24h totals, two simple disaggregation approaches of the daily precipitation time series were assessed.

The first technique was a proportional distribution, where the observed daily precipitations were considered constant during
the measuring period (6:00 UTC to 6:00 UTC the next day). Proportional parts of the original daily time series were allocated into a new moving average of 24h totals (Fig. 3).

The second approach aimed at getting closer to the chronology of the actual precipitation by relying on some informative variables during the reconstruction procedure. Data from the NCEP/NCAR reanalysis 1 (Sect. 2.1) were used for this purpose, despite their rough resolution (2.5°). Precipitable water and relative humidity (at 1000 hPa, 925 hPa or 850 hPa) were assessed
at the four points surrounding the catchment, and a time lapse between both series was allowed to take into account the significant distance separating the weather stations and the reanalysis grid point. The best proxy variable was identified by means of a correlation analyses (on non-zero values) with the 6-hourly precipitation time series over the period 1982–2007. Once the best proxy had been selected, its temporal profile was used in order to disaggregate the daily precipitation time series.

## 2.6 Calibration of the analogue method

The semi-automatic sequential procedure elaborated by Bontron (2004) was used to calibrate the AM (see also Radanovics et al., 2013; Ben Daoud et al., 2016). The analogy levels (e.g. the atmospheric circulation or moisture index) are calibrated sequentially. The parameters calibrated by this approach, for every level of analogy, are the spatial windows on which the predictors are compared and the number of analogues. The procedure, as defined by Bontron (2004), consists of the following steps:

1. Manual selection of the following parameters:

    (a) meteorological variable (e.g. geopotential height, temperature, relative humidity, etc.),

    (b) pressure level (e.g. 500 hPa, ...),

    (c) temporal window (hour of the day – e.g. 6:00 or 12:00 UTC),

    (d) initial analogue numbers (e.g. $N_1 = 50$).

2. For every level of analogy:

    (a) Identification of the most skilled unitary cell (one point for moisture variables and four for geopotential heights when using the S1 criterion) over a large domain. Every point (or cell) of the full domain is assessed on all predictors of the level of analogy, jointly (consisting generally of the same variable but on different pressure levels and at different hours).

    (b) From this most skilled point, the spatial window is expanded by successive iterations in the direction of greater performance gain. The spatial window grows by repeating the previous steps until no improvement is reached.

    (c) The number of analogue situations $N_1$ is then reconsidered and optimized for the current level of analogy.

3. A new level of analogy can then be added, based on other variables (such as the moisture index) at chosen pressure levels and hours of the day. The number of analogues for the next level of analogy, $N_2$, is initiated at a chosen value. The procedure starts again from step 2 (calibration of the spatial window and the number of analogues) for the new level. The parameters calibrated on the previous analogue levels are fixed and do not change.

4. Finally, the numbers of analogues $N_1$ and $N_2$ for the different levels of analogy are reassessed. This is done iteratively by varying the number of analogues of each level in a systematic way.

The calibration is done in successive steps with a limited number of parameters. Previously calibrated parameters are generally not reassessed (except for the number of analogues). More advanced techniques, such as using genetic algorithms (Horton et al., 2016), exist but are out of the scope of the present study.

## 3 Results

### 3.1 Consequences of the archive reduction

The performance loss resulting from the reduction of a 47 year archive (1961–2008, corresponding to the daily time series) to 25 years (1982–2007, corresponding to the hourly time series in Sect. 2.4) was assessed with the original method without

MTW. Both 2Z (Table 1) and 2Z-2MI (Table 2) methods were considered. The AM parameters were calibrated on the original archive (see resulting analogue numbers in Table 3) and were used thereafter.

The impact of the archive reduction can be seen in Fig. 4 for the 2Z method and in Fig. 5 for the 2Z-2MI method. As expected, a loss of performance was observed for each station, except for that of Aigle, which seemed relatively indifferent to this change. This loss was globally significant, with up to –1.89 points (% of CRPSS) for Visp and the 2Z method.

### 3.2 Influence of the MTW on the analogy criterion

#### 3.2.1 Changes in the atmospheric circulation analogy

When searching for analogues in the first level of analogy, such as on the geopotential heights in the 2Z method (Sect. 2.2), there are 4 times as many candidates (even though not fully independent) with the MTW than before (Sect. 2.4). Figure 6 shows the changes in the distributions of the analogy criterion (S1) for the $1^{st}$, $5^{th}$, $20^{th}$, and $40^{th}$ analogue rank at the Ulrichen station

over the whole calibration period (1961–2008 – the full period could be used here, as precipitation was not considered at this stage), due to introduction of the MTW. The shape of the distributions of the conventional approach and the MTW were found to be similar, but the values of the analogy criterion were reduced (shifted to the left) and were, therefore, better. For analogues with higher ranks (e.g. $20^{th}$ or $40^{th}$), the difference between the two distributions was larger than for the first ranks, which means that the improvement increased with the rank of the analogues.

The improvements of the analogy with the rank of the analogues are summarized in Fig. 7, which shows (top) quantiles of the S1 criterion for the conventional method and the MTW, and (bottom) quantiles of the relative reduction (meaning improvement) due to the MTW. This confirms that all quantiles were similarly reduced (S1 distributions keep their shape), and that this improvement was constantly increasing from the first to the last analogue (Fig. 7 bottom).

#### 3.2.2 Changes per precipitation class

The impact of the MTW on the analogy criterion has been analysed per precipitation classes (for the target day). The results are summarised in Fig. 8 by the median reduction of S1 for the days with precipitation (organized into classes) between two thresholds. With the number of cases per class being reduced, the curves are not as smooth as in previous analyses. It is nevertheless clear that the improvements were larger for days with higher precipitation. Once again, the results for all the other stations were similar.

### 3.2.3 Changes per season

Atmospheric dynamics vary greatly from one season to another, reflecting on the performance of the AM that is generally lower between June and August (Bliefernicht, 2010). The effect of the MTW on the S1 criterion per season is presented in Fig. 9. Differences between the seasons were substantial, with greater improvements for winter than summer.

### 3.2.4 Changes in the moisture analogy

When adding the second level of analogy of the 2Z-2MI method (Table 2), the number of candidate situations did not increase, as they were conditioned by the $N_1$ previously selected analogues, but their dates had changed. In contrast to the AM on the atmospheric circulation only, both a reduction or an increase of the RMSE analogy criterion values were possible with the MTW compared to the static approach. Indeed, Fig. 10 shows an almost insignificant improvement of the RMSE values. Unlike the first level of analogy, the relative changes of the RMSE values were distributed relatively symmetrically around zero, with improvements and losses of the same amplitude.

### 3.3 Impact of the MTW on prediction skill

The new performance scores (CRPSS) of the precipitation prediction are provided in Fig. 4 and 5 for the 2Z and 2Z-2MI methods, respectively, for the reduced archive (1982–2007). The gains relative to the static approach on the same archive ranged from 0.57 to 2.14 points (% of CRPSS) for the 2Z method and from 1.53 to 2.20 points for the 2Z-2MI method.

#### 3.3.1 Improvement per precipitation classes

The S1 criterion was previously found to show greater improvement for higher precipitation values (Sect. 3.2.2). The changes in terms of prediction skill were also assessed for the precipitation classes. Fig. 11 synthesizes these differences for the Ulrichen station, with the other stations having the same behaviour. The performance score was improved for days with higher precipitation after the introduction of the MTW. In contrast, for non-rainy days and small precipitation values, the performance scores were not improved.

#### 3.3.2 Recalibration of AM parameters

The previous assessment of the performance improvement was established with the original parameters optimized with a fixed time window. However, one can assume that the introduction of the MTW might change the optimum value of some parameters. The calibration (see Sect. 2.6) should then be reprocessed.

After recalibrating, changes in the AM parameters could be observed for both the 2Z and 2Z-2MI methods. Among these, the zonal extent of the spatial windows of the circulation analogy decreased slightly (not shown). More significantly, the optimal number of analogues $N_1$ and $N_2$ increased after introducing the MTW by a considerable magnitude (Table 3): 25 % to 83 % for the 2Z method and 20 % to 67 % for the final selection of the 2Z-2MI method. The number of analogues $N_1$ of the first

analogy level of the 2Z-2MI method reached three times its previous value for the Visp station ($N_1$ = 135 instead of 45; Table 3).

The values of the CRPSS scores for both methods (Fig. 4 and 5) have significantly increased after recalibration. When analysing the change in performance per precipitation class, the results (not shown) were very similar to the observations in
Sect. 3.3.1, with a slight performance increase for small precipitation values that can be observed at the expense of higher amounts, due to the higher number of analogues.

## 3.4 Using a reconstructed archive

The reconstructed time series cover the full period. However, they were first assessed using the MTW on the reduced period in order be comparable with the real sub-daily precipitation time series, and thus to evaluate their relevance and the possible loss
in performance score.

When using the first reconstructed time series based on the proportional distribution (see Sect. 2.5), the AM performance was degraded and was even below the conventional method without MTW (Table 4 to compare to Fig. 4 and 5).

For the second approach, the synoptic proxy for the temporal disaggregation of the daily time series had to be identified first. The results are illustrated this time for the Zermatt station. Among the considered moisture variables (see Sect. 2.5), the best
proxy was the precipitable water at 45° N and 7.5° E, with a time offset of –6 h (Table 5). Table 6 presents the CRPSS scores obtained by the disaggregated series using the proxy of precipitable water which are also lower than the conventional method.

## 4 Discussion

### 4.1 Improvement of the selection of analogue situations

For the first level of analogy, the improvement of the S1 criterion (Fig. 7 bottom) started approximately at 5 % for the first
analogue and reached more than 10 % for the last ($40^{th}$) (Sect. 3.2.1). This increase in improvement with the analogue rank can be explained by the accumulation of better analogues in the selection, with new better situations pushing previous analogues to higher ranks. The curve representing the minimum improvement is mostly superior to zero, meaning that the criteria have been improved on most analogue ranks for every day of the calibration period. All other stations had a similar improvement in the S1 criterion, both in terms of the distribution shape and amplitude. The introduction of the MTW allows finding better
analogue situations in the first level of analogy, resulting in a selection of days with better similarity in atmospheric circulation.

When adding a second level of analogy with moisture variables, the criteria values (RMSE) were not improved by the MTW (Sect. 3.2.4), as the number of candidates was not higher (still the $N_1$ days selected in the first level of analogy). However, this result of a globally null improvement of the RMSE values does not mean that the 2Z-2MI method cannot be improved by the MTW. It means that after the selection of the analogue situations for the synoptic circulation, the new candidate dates did not
provide better analogues in terms of moisture. However, the selected dates have changed in the first level of analogy and also

in the final selection, and thus the distributions of precipitation values were different, which did improve the prediction skill (Sect. 3.3).

The introduction of the MTW improved the selection of synoptic analogues. Independently of its impact on the prediction skill for precipitation, or the availability of a predictand time series with sub-daily time step, this improvement has a potential in itself for application on long meteorological archives. For example, when processing forecasts for a target day showing synoptic similarity with situations from the past related to extreme weather, even if for them no precipitation archive is available. Indeed, some of those days with strong precipitation events may be documented, either qualitatively in the daily press or more quantitatively in flood reports. It is thus worth to known that the situation at hand has had such analogue in the past. Another possible application is a quality assessment of the selection of analogues on a shorter period, where precipitation data with a high temporal resolution is available. Indeed, if the selection of analogues with the MTW on the long period, for a specific target day, differ from the selection on the short period, this may point out a sub-optimal forecast. Finally, other predictands might not need sub-daily total values, but point observations (e.g. hail, or extreme wind gusts), which make them easier to use with the MTW.

## 4.2  Improvement of precipitation prediction

The MTW was found to improve the skill of the precipitation prediction for both the 2Z and 2Z-2MI methods (Fig. 4 and 5, Sect. 3.3). Moreover, it required no additional predictor. With the introduction of the MTW, the performance loss related to the reduction of the archive length was negated (considering a 25 year archive of sub-daily data; the difference could be higher with a shorter archive). It was in this case of the same magnitude as if the length of the archive doubled. Note, that despite the number of candidate situations being 4 times as many, the gains seem to be lower than for a quadrupled archive length. The likely reason is that an actual longer archive would contain more atmospheric situations that might have been observed less frequently during a shorter period. Moreover, high correlations between sub-daily circulation patterns are expected. The MTW, therefore, did not create 4 times as many independent data, but the increase in information was nevertheless substantial. In this regards, a reanalysis dataset with a higher temporal resolution might not improve the performance significantly more than a 6-hourly dataset. In contrast, reducing the MTW to a 12-hour time step might reduce redundancy in the archive, but might also reduce the chance of finding better analogue situations, as circulation patterns can evolve substantially in 12 hours.

Moreover, in a transient climate, an eventual nonstationarity of the link between predictors and precipitation might discard the relevance of analogues from the distant past and increase the relevance of using a more recent and shorter archive rather than a long one. In such cases, the archive inflation brought by the MTW is also relevant.

Both the analogy criteria (Sect. 3.2.2) and the performance scores (Sect. 3.3.1) were improved to a greater extent for days with heavier precipitation. This is likely due to the fact that higher precipitation events are a consequence of atmospheric conditions with greater dynamics, such as weather disturbances, which have a well-marked temporal evolution. Indeed, the position of the driving elements, such as the low pressure centres and the fronts, change significantly during a day. These situations are less numerous than anticyclonic situations, which makes it less likely to find very good analogues at the same time of the day. We can, therefore, expect to improve these situations with greater dynamics more significantly when introducing a

MTW, as better matches to the target situation may be found. In contrast, days with low dynamics in the atmospheric circulation, such as anticyclonic situations, will not be radically improved by the introduction of the MTW.

The MTW improved the prediction for days with heavier precipitation, and should improve the prediction of extremes due to better analogue situations, but also due to possible new extreme values resulting from 24h totals with a certain time shift.

However, even though the distribution of analogue precipitation values should move towards the targeted extreme, providing a better prediction, the MTW itself does not allow to predict extreme events that were not yet observed and are, therefore, not present in the archive. The extremes in AMs can be modelled by, for example, extrapolation of a truncated exponential or gamma distribution fitted to the analogue values (Obled et al., 2002). Another possible approach is by combining AMs with other methods (e.g. Chardon et al., 2014). From this perspective, the MTW might improve the prediction of extremes as it

improves the distribution of precipitation values for days with higher precipitation, on which post-treatment techniques rely. However, this goes beyond the scope of the present study.

### 4.3 Seasonal effect

Section 3.2.3 and Fig. 9 revealed a difference in the improvement of the S1 criterion according to the season, with greater improvements for winter than summer. One hypothesis is that the diurnal effects of the summer months have an influence on

the atmospheric circulation at least in the lower layers. This effect is in phase with the daily cycle, and good analogues are essentially found for the same hours.

An analysis of the selected hours for the geopotential height predictor seems to confirm this assumption (Fig. 12). It was found that the new choice of the temporal window in winter, when using the MTW approach, is well balanced between the four options. This means a change of 75 % of the analogues selected compared to the conventional approach. On the contrary,

the days during the summer months had a preference for the initial temporal window (Z500 24 h & Z1000 12 h), likely due to more pronounced diurnal effects, which reduced the potential for improvement of the criteria. The other seasons were between these two extremes, which is consistent with their respective improvements. This seasonal effect was observed for each station in a very similar way and even with slightly larger amplitude than for Ulrichen.

### 4.4 On the increase in the number of analogues

After the recalibration of the AMs with the MTW, the optimal analogue numbers were significantly higher than in the original methods (see Sect. 3.3.2 and Table 3). As shown in Fig. 7, the improvement of the S1 criterion grew along with the rank of the analogue, which shows an accumulation of better analogue situations in the distributions. It seems profitable to widen the selection of analogues in order to also keep some whose rank has increased, as they appear to be relevant to the prediction of the precipitation values. The number of good analogues was globally increased.

A higher number of analogues generally means, with an archive of fixed length, a poorer analogy. Indeed, when the choice of the predictors or the parameters are improved, leading to a better prediction, the optimal number of analogue situations decreases. However, when the length of the archive increases, the optimal number of analogues increases too for a better performance up to a certain threshold (Bontron, 2004; Hamill et al., 2006). The observed increase in the number of analogue

situations with the MTW resulted in better performance skills for the given methods, as it can be seen as an inflation of the archive. However, if new relevant predictors were added to the method, the number of analogues would then decrease.

## 4.5 Why not just make 6-hourly predictions?

One can question the interest of using moving daily totals when 6-hourly precipitation series can be predicted instead. However, the 6-hourly time series generated by the AM might not represent accurately the intra-daily precipitation distribution (results not shown). In addition, sometimes a resolution finer than the daily time step is not needed and another disaggregation technique may be used afterwards. Finally, when using a reconstructed precipitation archive, the errors in intra-daily precipitation distributions have a smaller impact than 24h daily totals.

## 4.6 On the use of an old reanalysis dataset

The considered reanalysis dataset, which is the NCEP/NCAR reanalysis 1 (Kalnay et al., 1996), is relatively old and has a coarse resolution (2.5°). Newer reanalysis datasets could have been used in this study and might have resulted in higher performance scores. However, Ben Daoud et al. (2009) showed that the sensitivity of the method to the reanalysis dataset is rather small. This is particularly true for simple AMs that mainly rely on the atmospheric circulation, which is already well defined at a coarser resolution. That might not be the case in more elaborate methods relying on thermodynamic data. A comparative analysis of several reanalysis datasets within the AM is being conducted and will be the topic of a dedicated study.

Moreover, one can assume that the gain observed here by introducing a MTW should also be found for a better reanalysis dataset. Indeed, a better dataset does not negatively influence the fact that one can find better analogue situations at other hours of the day rather than at fixed hours.

## 4.7 Relevance of the reconstructed precipitation archives

The two simple methods assessed to reconstruct precipitation archives (Sect. 2.5) did not result in valuable outputs (Sect. 3.4). Indeed, the performance improvement brought by the MTW was lost due to the poor quality of the precipitation archives. A slight improvement could be obtained for the second method relying on a proxy variable compared to the proportional distribution method, but it was still relatively small, and most of the benefit of the MTW was lost. A more recent reanalysis archive with more accurate moisture variables might produce better proxies.

These attempts to transpose the MTW on the total archive highlighted the importance of the actual rainfall chronology. The MTW is beneficial, provided that the precipitation series are close to the observed one. Without a precipitation series with an accurate sub-daily chronology, the introduction of a MTW does not improve the precipitation prediction.

# 5 Conclusions

The AMs are most often based on a daily time step due to the availability of long precipitation archives. However, it is unlikely that two analogue synoptic situations, that evolve relatively quickly, would correspond optimally at the same hours of the day. It is probable that better matches can be found at another time, which can change the selection of the analogue dates.

As Finet et al. (2008) had previously shown, the introduction of a MTW allows finding better analogue situations in terms of the atmospheric circulation. It has been demonstrated in this study that the improvement of the S1 criterion values was growing with the rank of the analogue. This was due to the accumulation of better analogues within the predicted distributions.

The improvement of the circulation analogy was found to be more important for days with heavier precipitation, which are generally related to more dynamic atmospheric situations and are less frequent in the archive. These situations have more specific circulation patterns that are evolving more rapidly. Therefore, a MTW was found to be of particular interest in this kind of situation, benefiting the prediction of heavier precipitation events.

A seasonal effect has been highlighted, as the MTW was more profitable for winter than summer. The reason is likely that the diurnal cycle has a bigger effect in summer than in winter which results in better analogues at the same time of the day. The preference for the same hours in summer has been demonstrated; whereas, 75 % of the analogue situations were selected at a different time in winter.

The impact of the MTW on the prediction skill was never assessed before due to the shortcoming of long precipitation series at a sub-daily time step. Here, it was assessed for a 25 year time series with a high temporal resolution. After the introduction of the MTW, the performance scores increased of the same magnitude as if the length of the archive doubled.

The parameters were then calibrated again, using the MTW. Some parameters changed, with the main difference being the number of analogues, which systematically and significantly increased compared to the original set. A wider selection of analogue situations, containing those whose rank decreased, seemed beneficial for the prediction performance. The number of good analogues was globally increased in the same way as if the archive length increased. This change seems to benefit the prediction of days with small precipitation totals.

The importance of the quality of the precipitation archive was also demonstrated, as simplistic reconstructions of 6-hourly time series led to a loss of all the improvement brought by the MTW. The precipitation prediction is improved only when the precipitation chronology is close to the accurate one. Attempts to reconstruct longer time series based on simplistic proportional distributions or by using meteorological variables from the NCEP reanalysis 1 dataset as proxy did not succeed. Other reanalysis datasets with more accurate moisture variables could eventually perform better.

The use of the MTW relies partly on the availability of long precipitation series at a sub-daily time step and with high accuracy. First, these archives of high temporal resolution precipitation data are increasing over time. Another possible source of such archives is the establishment of precipitation reanalysis at a regional scale or the use of reanalysis-driven regional climate models or limited area models over a long period. Even though outputs from these models might be biased or not accurate enough, information regarding the timing of the precipitation events could be useful in disaggregating the station time series.

Finally, since long meteorological archives (reanalysis datasets) are more and more available, the improvements proposed by the MTW especially for days with heavy precipitation may be interesting even without long continuous precipitation archives. For example, recent target day may have synoptic similarity with situations from the early twentieth century, for which no continuous daily precipitation archive is available. However, some of those days with strong precipitation events may be documented, either qualitatively in the daily press or more quantitatively in flood reports. Nevertheless, it is worth to known that the situation at hand has had such analogue in the far past.

The use of the MTW in the AM can already be considered now for several applications in different contexts, may it be for operational forecasting or climate-related studies.

## Appendix A: Acronyms

2Z     Name of the analogue method of the atmospheric circulation

2Z-2MI     Name of the analogue method composed of a first level on the atmospheric circulation and a second level on a moisture index

AM     Analogue method

CRPS     Continuous ranked probability score

CRPSS     Continuous ranked probability skill score

MTW     Moving time window

NCAR     National Center for Atmospheric Research

NCEP     National Center for Environmental Prediction

RH850     Relative humidity at 850 hPa

RMSE     Root-mean-square error

S1     Teweles and Wobus (1954) score

TPW     Total precipitable water

Z1000     Geopotential height at 1000 hPa

Z500     Geopotential height at 500 hPa

*Competing interests.* The authors declare that they have no conflict of interest.

*Acknowledgements.* We thank the Swiss Federal Office for Environment (FOEV), the Roads and Water Courses Service, Energy and Water Power Service of the Wallis Canton, and the Water, Land, and Sanitation Service of the Vaud Canton which financed the MINERVE (Modélisation des Intempéries de Nature Extrême des Rivières Valaisannes et de leurs Effets) project that started this research. Thanks to Dominique Bérod for his support.

5    The fruitful collaboration with the Laboratoire d'Etude des Transferts en Hydrologie et Environnement of the Grenoble Institute of Technology (G-INP) was made possible thanks to the Herbette Foundation.

NCEP reanalysis data provided by the NOAA/OAR/ESRL PSD, Boulder, Colorado, USA, from their Web site at http://www.esrl.noaa.gov/psd/. Precipitation time series provided by MeteoSwiss.

The authors would also like to acknowledge the work of anonymous reviewers that contributed to improving this manuscript.

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

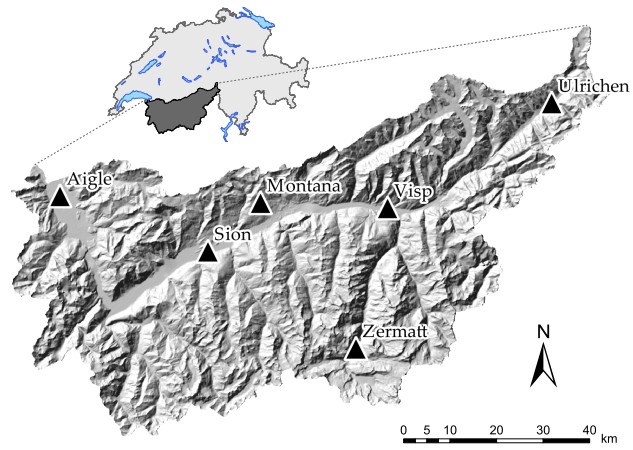

**Figure 1.** Position of the six weather stations of interest (Ulrichen, Zermatt, Visp, Montana, Sion, and Aigle) in the upper Rhône catchment in Switzerland.

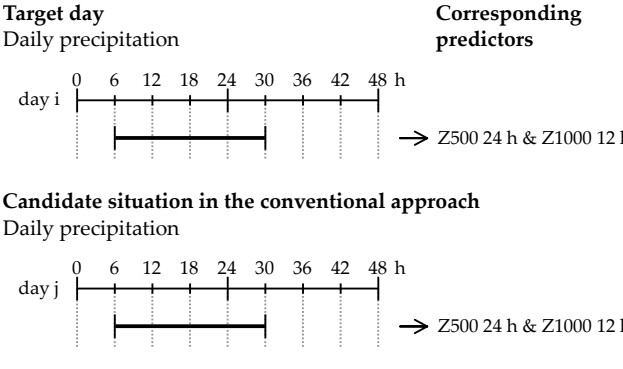

**Figure 2.** Illustration of the principle of a moving time window. The target situation is the same for the conventional approach and the MTW, while the candidate situations are 4 times as many with the MTW. The larger horizontal bars represent the 24h precipitation totals.

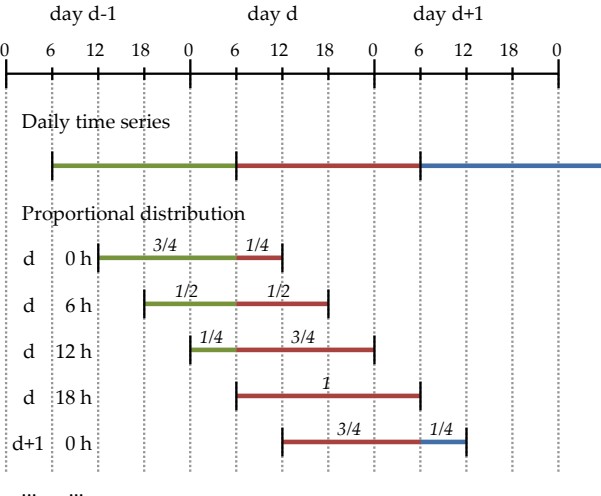

**Figure 3.** Illustration of the generation of 24h total moving averages by means of a proportional distribution. The colours refer to the corresponding day of the daily time series.

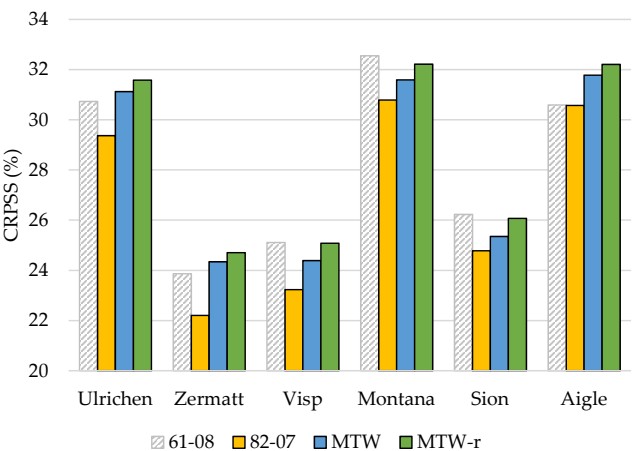

**Figure 4.** Performance score (CRPSS) of the AM on the atmospheric circulation at the different stations for (dashed) the full archive, i.e. 1961–2008; (yellow) the reduced archive, i.e. 1982–2007; (blue) the introduction of the MTW on the reduced archive; and (green) the recalibrated parameters of the AM with the MTW.

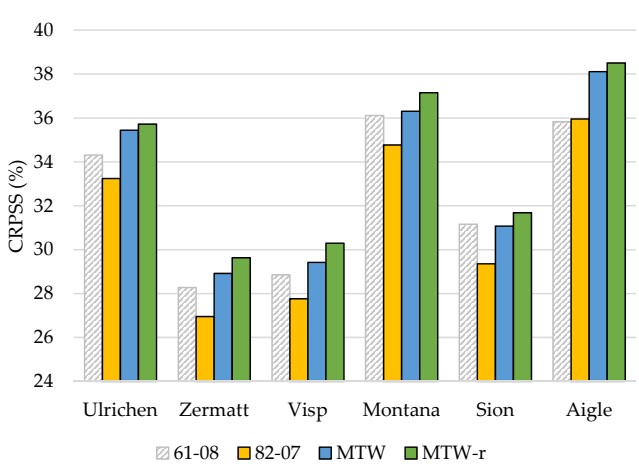

**Figure 5.** Same as Figure 4, but for the analogue method with a second level with the moisture variables.

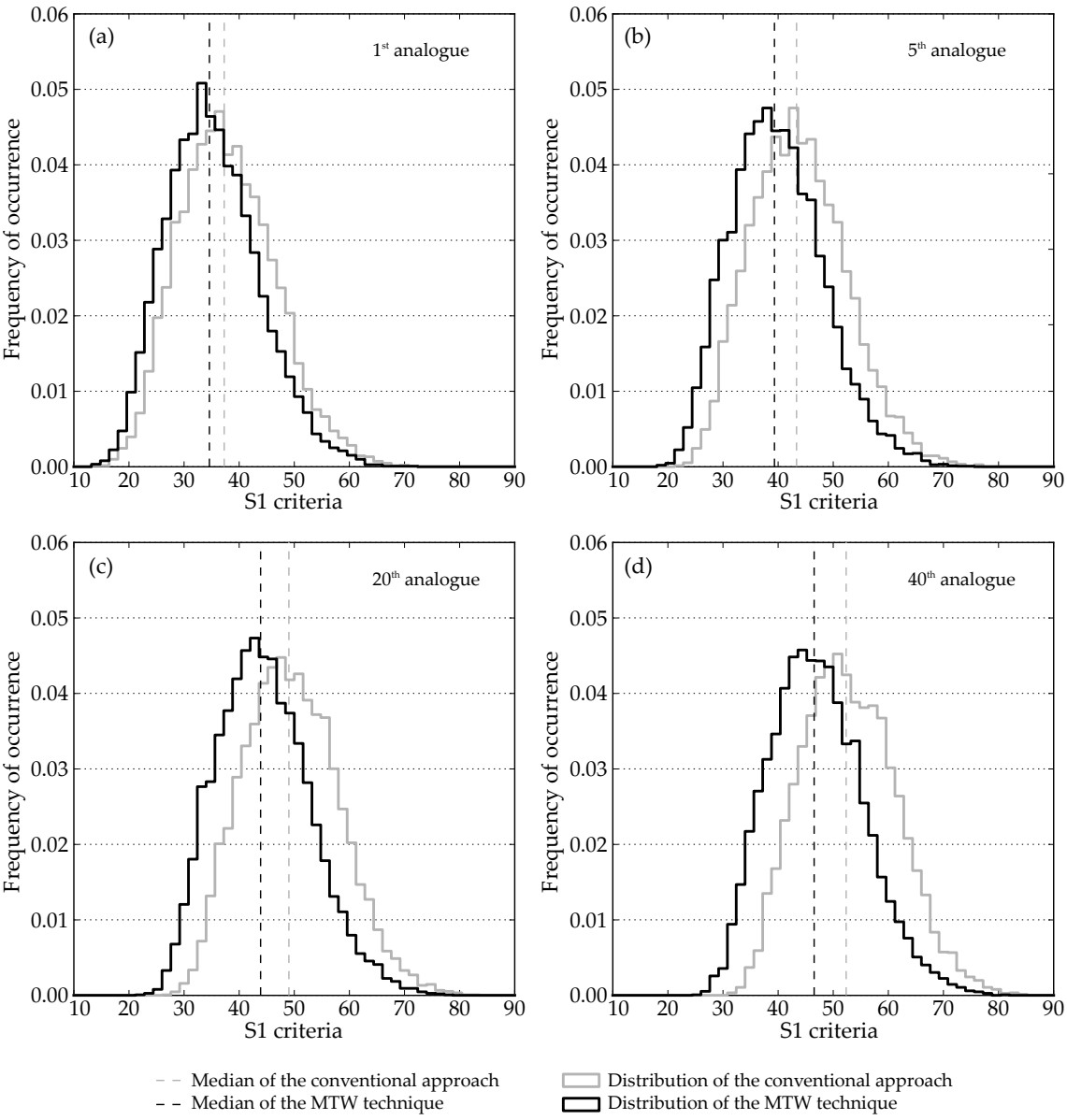

**Figure 6.** Changes in the S1 criterion distributions due to the MTW of (a) the $1^{st}$, (b) $5^{th}$, (c) $20^{th}$, and (d) $40^{th}$ analogue ranks for the Ulrichen station over the whole calibration period (1961–2008).

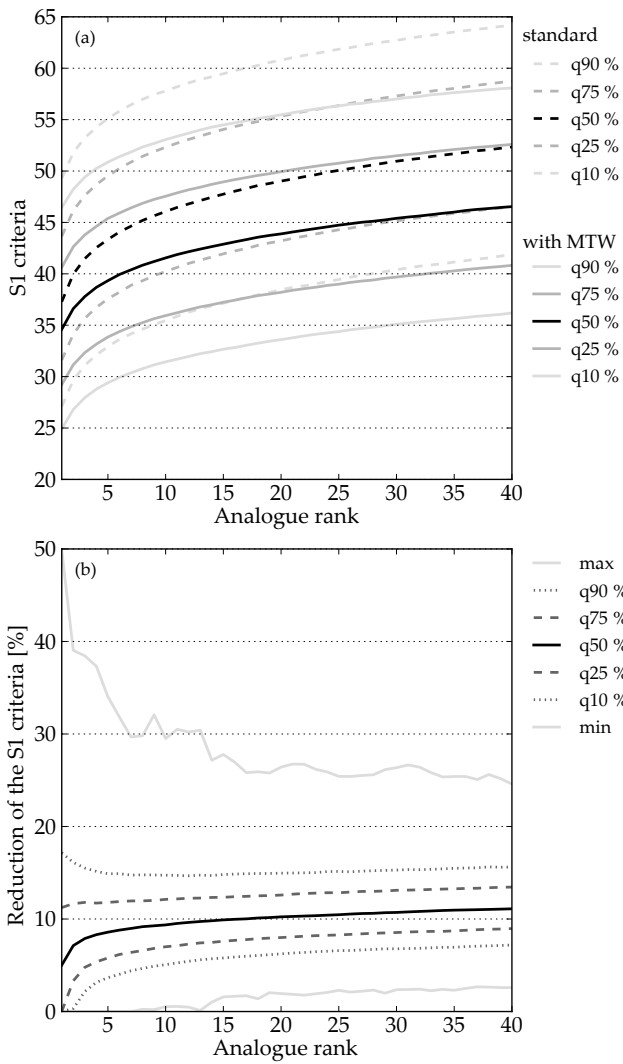

**Figure 7.** Synthesis of the changes in the S1 criterion due to the MTW for the Ulrichen station depending on the rank of the analogue. (a) Quantiles of the S1 distributions with and without the MTW. (b) Quantiles of the relative improvements of the S1 criterion when using the MTW.

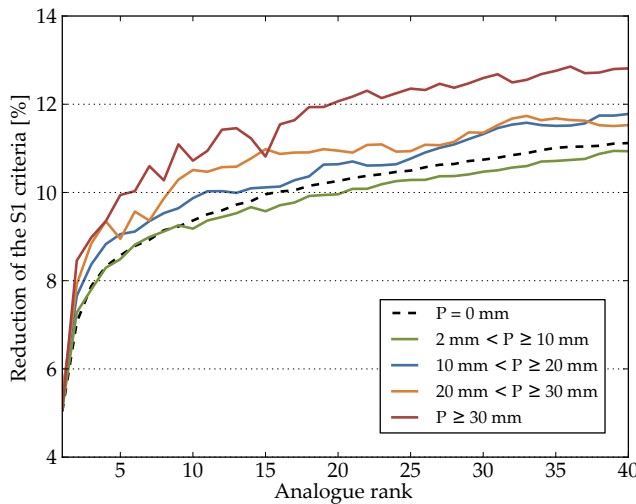

**Figure 8.** Distribution of the median improvements of the S1 criterion due to the MTW depending on daily precipitation thresholds at the Ulrichen station.

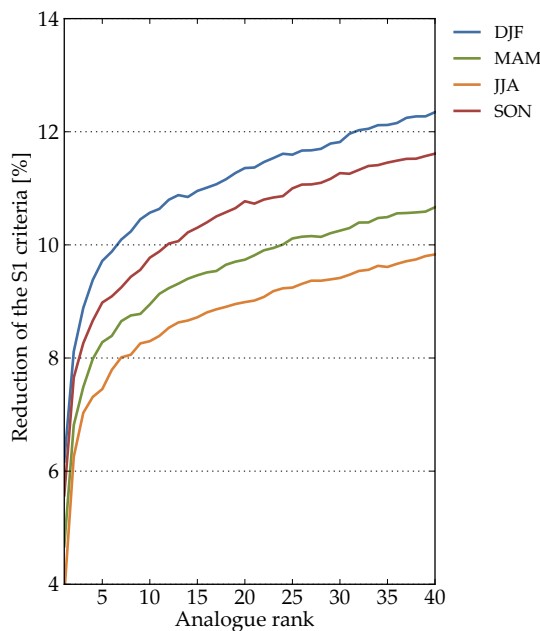

**Figure 9.** Seasonal effect on the median reduction of the S1 criterion for the Ulrichen station due to the MTW. DJF: winter, MAM: spring, JJA: summer, and SON: fall.

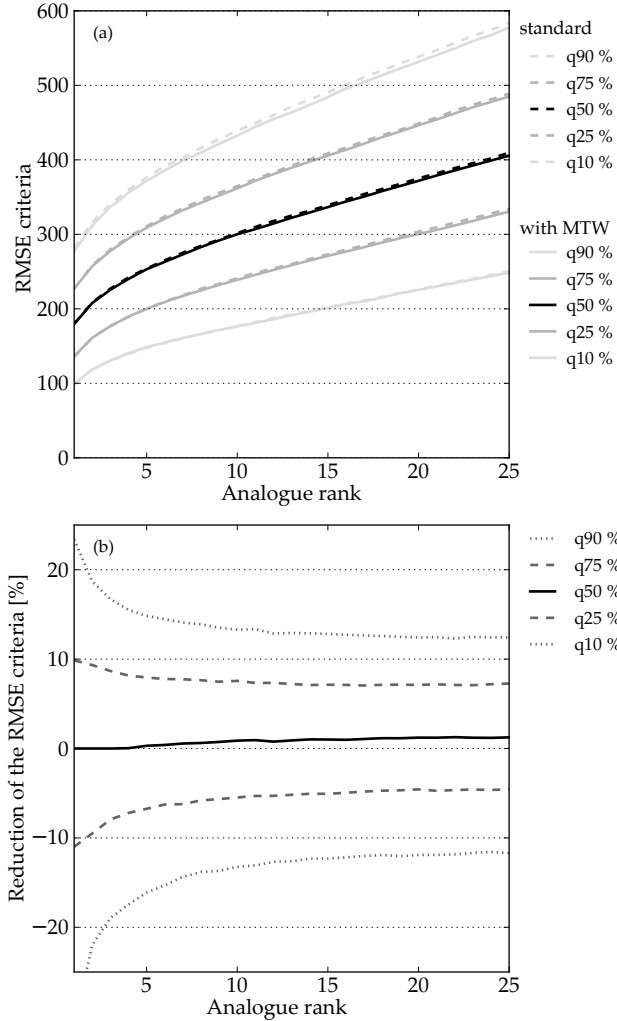

**Figure 10.** Synthesis of the changes in the RMSE criterion due to the MTW for the Ulrichen station depending on the rank of the analogue. (a) Quantiles of the RMSE distributions with and without the MTW. (b) Quantiles of the relative improvements of the RMSE criterion when using the MTW.

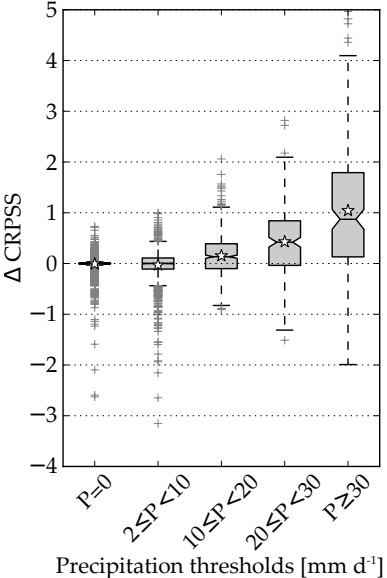

**Figure 11.** Differences of the CRPSS performance score due to the introduction of the MTW as a function of daily precipitation thresholds at the Ulrichen station. The stars represent averages.

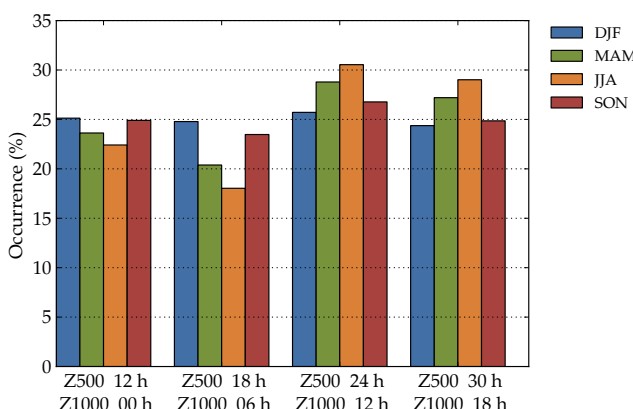

**Figure 12.** Distribution of the predictor hours on the selected analogue dates, when using the MTW, depending on the season for the Ulrichen station.

**Table 1.** Parameters for the reference method on the atmospheric circulation (2Z). The first column is the level of analogy (0 for preselection), the second column is the meteorological variable, and then its hour of observation (temporal window). The criterion used for the current level of analogy is then provided, as well as the number of analogues.

| Level | Variable | Hour | Criterion | Nb |
|-------|----------|------|-----------|-----|
| 0 | $\pm 60$ days around the target date | | | |
| 1 | Z1000 | 12 h | S1 | $N_1$ |
| | Z500 | 24 h | | |

**Table 2.** Parameters of the reference method with moisture variables (2Z-2MI). Same conventions as Table 1

| Level | Variable | Hour | Criterion | Nb |
|-------|----------|------|-----------|-----|
| 0 | $\pm 60$ days around the target date | | | |
| 1 | Z1000 | 12 h | S1 | $N_1$ |
| | Z500 | 24 h | | |
| 2 | TPW * RH850 | 12 h | RMSE | $N_2$ |
| | TPW * RH850 | 24 h | | |

**Table 3.** Optimal number of analogues (of the first and second level of analogy, respectively, $N_1$ and $N_2$) of the method based on the atmospheric circulation only (method 2Z) and the method with a second level of analogy with moisture variables (2Z-2MI) on the full archive (Standard), and after recalibration using the MTW (MTW-r).

| Station | Standard | | | MTW-r | | |
|---------|----------|----------|-------|-------|----------|-------|
| | 2Z | 2Z-2MI | | 2Z | 2Z-2MI | |
| | $N_1$ | $N_1$ | $N_2$ | $N_1$ | $N_1$ | $N_2$ |
| Ulrichen | 40 | 60 | 25 | 50 | 110 | 35 |
| Zermatt | 35 | 55 | 25 | 55 | 80 | 30 |
| Visp | 30 | 45 | 25 | 55 | 135 | 35 |
| Montana | 40 | 55 | 30 | 55 | 110 | 40 |
| Sion | 40 | 90 | 30 | 55 | 140 | 50 |
| Aigle | 50 | 100 | 35 | 75 | 135 | 45 |

**Table 4.** Values of the CRPSS (%) score for the original and the recalibrated parameters (with the sequential method, as described in Sect. 2.6) using the MTW approach on the disaggregated precipitation time series (short period) by means of the proportional distribution.

| Station | 2Z | | 2Z-2MI | |
| --- | --- | --- | --- | --- |
| | original | recalib. | original | recalib. |
| Ulrichen | 29.13 | 29.61 | 33.15 | 33.45 |
| Zermatt | 22.17 | 22.80 | 26.72 | 27.43 |
| Visp | 22.32 | 22.89 | 27.01 | 28.04 |
| Montana | 29.41 | 30.24 | 33.83 | 34.55 |
| Sion | 22.98 | 23.41 | 28.57 | 29.15 |
| Aigle | 29.07 | 29.46 | 34.66 | 35.09 |

**Table 5.** Value of the coefficient of determination between the reconstructed 6-hourly precipitation time series using the listed variables and the accurate time series for the period 1982–2007. The grid points surrounding the region are: 1) 5° E, 47.5° N; 2) 5° E, 45° N; 3) 7.5° E, 47.5° N; and 4) 7.5° E, 45° N. The highest coefficient of determination is indicated in bold.

| Variable | Point | Time lapse | | | | |
| --- | --- | --- | --- | --- | --- | --- |
| | | −12 h | −6 h | 0 h | +6 h | +12 h |
| RH1000 | 1 | 0.668 | 0.669 | 0.684 | 0.683 | 0.670 |
| | 2 | 0.669 | 0.669 | 0.683 | 0.681 | 0.669 |
| | 3 | 0.662 | 0.673 | 0.691 | 0.682 | 0.673 |
| | 4 | 0.666 | 0.671 | 0.688 | 0.681 | 0.668 |
| RH925 | 1 | 0.672 | 0.673 | 0.684 | 0.684 | 0.675 |
| | 2 | 0.674 | 0.674 | 0.683 | 0.682 | 0.672 |
| | 3 | 0.662 | 0.673 | 0.691 | 0.682 | 0.673 |
| | 4 | 0.666 | 0.671 | 0.689 | 0.681 | 0.668 |
| RH850 | 1 | 0.675 | 0.675 | 0.679 | 0.678 | 0.671 |
| | 2 | 0.681 | 0.690 | 0.691 | 0.677 | 0.664 |
| | 3 | 0.665 | 0.680 | 0.693 | 0.683 | 0.675 |
| | 4 | 0.675 | 0.694 | 0.706 | 0.681 | 0.659 |
| TCW | 1 | 0.688 | 0.687 | 0.667 | 0.655 | 0.652 |
| | 2 | 0.697 | 0.699 | 0.669 | 0.644 | 0.644 |
| | 3 | 0.686 | 0.708 | 0.689 | 0.655 | 0.648 |
| | 4 | 0.696 | **0.721** | 0.696 | 0.643 | 0.636 |

**Table 6.** Values of the CRPSS (%) score for Zermatt for the original and the recalibrated parameters (with the sequential method, as described in Sect. 1) using the MTW approach on the disaggregated precipitation time series (short and long periods) by means of proxy variables from the reanalysis dataset.

| Period | 2Z | | 2Z-2MI | |
|---|---|---|---|---|
| | original | recalib. | original | recalib. |
| 1982–2007 | 22.57 | 23.14 | 27.11 | 27.71 |
| 1961–2008 | 23.81 | 24.38 | 28.42 | 28.86 |