# Peer review of "The analogue method for precipitation prediction: finding better analogue situations at a sub-daily time step"

_Hydrology and Earth System Sciences, 2016_

## Referee Comment (RC1) · Anonymous Referee #1 · 4 Jul 2016

General comment

In classical applications of the analog method, analog situations of some target day are identified in an archive of days defined on the same temporal window, namely 6H00 a.m. to 6h00 of the following day. This window is classically fixed to be close to the temporal window considered by meteorologists for the measurement of rainfall at daily gauges. A number of analog situations potentially also exist when other temporal windows are considered, e.g. 12am to 12 am. The paper explores the possibility to increase the prediction skill of atmospheric analogs using a Moving Temporal Windows. As atmospheric predictors are available at a 6hourly time step, 4 different atmospheric situations are thus available for each day instead of one (6h>6h, 12>12, 18>18, 24>24).

[Figure]

The authors show that when the same archive period is considered, the increased number of potential analogs (x4) for any target day leads for a significant improvement of the prediction skill.

They also discuss the issue of the length of the archive. The archive which has to be available, for both the predictor and the predictand, at a 6 hourly time step covers obviously a much smaller time period than classical archives of daily rainfall (roughly 2 times smaller in the present case). The authors show that the benefit of a MTW is reduced by the limited length of the high resolution archive.

The analysis is done for a couple of precipitation stations in South-Western Switzerland. It focuses on differences between the classical approach and the MTW one, describe how results depend on the "dynamism of the atmospheric situation" or on seasons, explore if a disaggregation of the longer lower resolution archive would be worth to improve prediction skill.

The paper gives a very original and valuable contribution to the issue of probabilistic weather prediction with the analog approach. It is globally sound and describes in a intelligible way different aspects of the problematic. Figures are clear as tables (perhaps too many tables). Some points are however not optimal and should be modified/clarified and or completed. They are listed below. For this reason, I would recommend a major revision of the manuscript.

Specific comments

Introduction.

To my opinion, the introduction is not as an introduction should be.

- 75% of the present introduction describes the analog model that will be used afterwords in the work (it starts from line 35 and goes to line 78! Additional details are given from l88 to 92). The detailed description of the method should belong to a "Data and Method" Section.

- Most references involve one or more co-authors of the present paper as if the authors were not aware of the great amount of work that was done in the past years (decade) with this method.

- The issue discussed in the manuscript goes beyond this "fixed/moving" window issue. It more generally consider influence of the size of the data set (including the length of the archive) on the prediction skill. I am not aware of works that explored this issue but references to those, if any, and associated results have to be presented in the introduction.

- The final objective of the work is to increase the prediction skill of the method. Other means are possible for this. They should be also mentioned.

To my opinion, the introduction has thus to be rewritten. The detailed description of the method goes after. References to other works worldwide involving the analogs for general concepts or aspects of some relevance for the present study have to be integrated as well as possible (alternative) ways to improve their skill.

MTW strategy vs longer archive.

This analysis poses the question of the length of the archive. The MTW allows having 2 times more analogs than a two times longer low resolution archive. It would have been interesting to put in perspective the present results with results of other works where this issue has been explored. Do you find similar gain here. We can imagine that the MTW strategy would not be as performant as a 4 times longer archive for two reasons :

1/ 2 overlapping situations of a given day (that beginning at 6h UTC and that beginning at 12h UTC o the same day) are likely to be highly correlated and may therefore contain more a less some redundant "large scale-small scale" information. The MTW strategy is thus not equivalent to a 4 times longer period at a daily time step. What would be the results if only the 2 temporal windows (6>6 and 18>18) (or for instance also (12>12

and 24>24)) are considered ? In such a case, the risk for redundancy is lowered and we could expect that this extended dataset (size is multiplied by two with respect to the initial daily one), provides a higher skill gain than the one obtained from the next archive extension (x2 to x4 as in your work). Could you precise what is the gain obtained from a MTWx2 and a MTVx4 approach ?

2/ An archive 4 times longer would possibly cover sligthly different climate contexts than those observed during any initial (shorter) period. For this reason, the longer archive is expected to also include atmospheric situations that are similar to the one of the target day, such similar situations being not in the initial shorter period for which high resolution archives are available

On the other hand, in a transient climate, the "large scale – small scale" link is likely to change and the predictions produced from analogs that are far back in the past are likely to be not really informative any more (or at least, not as informative as when they were used one or 2 decades ago). It could be therefore of some interest to keeping a smaller archive length to have similar [large-scale>small scale] configurations » the MTW is in this case also an alternative to extent the archive without relying on too old dates. ... These issues should be discussed in the manuscript

Number of analogs

Line 337 the authors say : "A wider selection of analog situations containing those whose rank decreased, seemed profitable for the prediction performance". This is somehow not what is expected. Previous studies show that the poorest the analog days are, the more analog days you have to consider sothat your prediction remain reliable (but you loose in resolution and in CRPS score). So, we could expect in the opposite case, that the better the analogs, the less analogs you need to make a good prediction and the better the prediction will be. This is obviously not the case from your results as you still find a better score in increasing the number of analogs. It could suggest a limit of predictability from the method. It clearly also suggests that important

auxiliary predictors are missed. Could you comment this.

Detailed Comments

71. Explain both terms of the decomposition. Another well-known decomposition of the CRPSS is that of Herbasch. Please mention it also and clarify the advantage of that of Bontron. It is not clear at this stage.

75. Why using a non conventional notation for the CRPS score. Please use the classical one (CRPSS) or justify

79. Clarify to what corresponds a CRPSS=1 or <0

93. "the first reason . . .". I do not understand this statement. Please clarify

110. A low resolution atmospheric reanalysis is used (NCEP/NCAR at 2.5°). What would be the increase in skill prediction with a higher quality reanalysis : resolution, model (interest of ERAint?). Would it be higher than the MTW strategy ?

105. Analog approach. It is not clear if all days are potential analog or if a seasonal stratification is considered for this identification. The results of your section 3.1.3 suggest that a seasonal stratification would be relevant (at least for summer). Could you comment ?

140. precise again the archive length for this section

146. "an increase in difference". I do not understand what kind of increase you comment. Please clarify

170. You use the mean intensity of rainfall as a proxy of "dynamism of the atmospheric situation". Another proxy could be the intensity of the variations within the geopotential fields (e.g. mean gradient value). Why did you not explore this ? It would better fit to your "dynamism" concept.

197. "In contrast to the earlier" : what is the reference AM you consider to state this

reduction / increase ?

208. STW > change for "analogy scores"

216. the differences ranges (delete the "s") : are they gain or losses. This is not clear at this stage. Please Clarify.

219. "moreover, it requires no additional predictor" > this is out of subject at this stage. Could be integrated / discussed in the discussion section. Next what is the gain that could be obtained from additional predictors. Should say a word on this point.

226. "the increasing positive trend of skill improvement" ???? should be "the increasing trend of skill ?". Please reword. I do not understand.

227. "is further improved" : when compared to what ?

230. "it seems as if the method. . ." > This sentence is likely not needed

246. has a slight effect" vs " have significantly increased" > it seems to be contradictory

251-253 : it is a conclusion of this section ?

259. section 3.2.3. why both sharpness and accuracy estimate should be comparable ? I do not see really the utility of this section. We do not know to what refers the accuracy / sharpness respectively, what we should expect as results. . . Not convincing.

279. Section 4.1. The disaggregation of daily prec. is an interesting strategy. Two simple approaches were considered in the present case without real success. What would be the perspectives of research for this issue ? This is probably missing in the present work. The disaggregation based on the proportional distribution is very not clear. What is the proportionality for ? I suspect that you assume that the rainfall intensity is constant throughout each day ? Is it right ? What is the length of the archive considered in this experiment ?

286 / 304. Clarify also the method used to make the disaggregation based on the

atmospheric proxy. What information is used ? how ? Do you scale each day the temporal profile of the proxy to disaggregate the daily precipitation amounts ?

347. perform instead of preform

349. "the first source of data..." The sentence is clumsy. Please reformulate. I do not see the link with "getting longer"

Figure 2 – in the figure / 3st graph > candidate "days" and not "day"

Figure 11 : - it is not clear for me. Does the recalibration lead to poorer scores ? - The two last plots are the same. > error

---

## Referee Comment (RC2) · Anonymous Referee #2 · 11 Jul 2016

This is a good paper, testing an advanced analogue-based approach for statistical downscaling. The proposed method, although largely based on an existing method, but with new advancements, is an innovative one. The results show that the added value of the method is not that high or as expected, but this is also an important finding and of importance for future studies.

The paper is well written; I advise moderate revision.

Major comments:

The most important disadvantage of any type of analogue or resampling method is its limitation in sampling extremes. The method does not allow to sample more extreme

conditions than the ones available in the historical time series. I suggest the authors add a comment on this and explain to what extent also their method is influenced by this limitation. For the same reason, it would be useful to explicitly investigate the accuracy of the tail of the frequency distribution for the extreme precipitation values, e.g. by plotting the precipitation value versus exceedance probability or return period.

Given that HESS is a journal on hydrology and earth sciences, I would be good to specify more on the benefits of the method for applications in hydrology and water resources. I also note that the literature review is very limited; with only few international journal papers cited. Both comments can be addressed by adding some more literature review in the introduction section on the needs and limitations of existing methods for statistical downscaling in hydrology. Given the focus of the paper on the sub-daily time step, I suggest to refer more on the needs in urban hydrology. Some references that can be cited to refer to these needs and limitations of current methods given below.

Minor comments:

Line 47: "the observation time of the predictor": it is not very clear what was meant by this. I assume it refers to the time of the day (as specific in line 45), but I suggest to clarify this better.

Maybe I overlooked, but I do not find details on how the sample sizes N1 and N2 are determined and optimized. I assume they are determined by optimizing the SCRP score. In Table 4, the N1 and N2 values therefore are shown next to the SCRP score. But how exactly the optimization is done (which optimization approach starting from which initial ranges for N1 and N2) is unclear. I suggest to add short clarification on this.

Line 74: change "following" to "follows"

Line 102: change "precipitations" to "precipitation"

Line 152: change "all quantile seem" to "all quantiles are"

Line 216: change "ranges" to "range"

Line 272 – Header of section 4: I would not use the word "attempts" . . .

Suggested references discussing the needs and limitations of statistical downscaling methods for urban hydrology:

Willems, P., Arnbjerg-Nielsen, K., Olsson, J., Nguyen, V.T.V. (2012), 'Climate change impact assessment on urban rainfall extremes and urban drainage: methods and shortcomings', Atmospheric Research, 103, 106-118

Arnbjerg-Nielsen, K., Willems, P., Olsson, J., Beecham, S., Pathirana, A., Bülow Gregersen, I., Madsen, H., Nguyen, V-T-V. 2013. Impacts of climate change on rainfall extremes and urban drainage systems: a review. Water Science and Technology, 68(1), 16-28

Sunyer, M.A., Bülow Gregersen, I., Rosbjerg, D., Madsen, H., Luchner, J., Arnbjerg-Nielsen, K. 2014. Comparison of different statistical downscaling methods to estimate changes in hourly extreme precipitation using RCM projections from ENSEMBLES. International Journal of Climatology, 35(9), 2528-2539

Willems, P. & Vrac, M. 2011. Statistical precipitation downscaling for small-scale hydrological impact investigations of climate change. Journal of Hydrology 402, 193–205.

Willems, P., Olsson, J., Arnbjerg-Nielsen, K., Beecham, S., Pathirana, A., Bülow Gregersen, I., Madsen, H., Nguyen, V-T-V. 2012. Impacts of climate change on rainfall extremes and urban drainage. IWA Publishing, 252p., Paperback Print ISBN 9781780401256; Ebook ISBN 9781780401263

Some other suggested references:

Zorita, E., Von Storch, H. 1999. The analog method as a simple statistical downscaling technique: comparison with more complicated methods. Journal of Climate, 12(8), 2474-2489.

[Figure]

Maraun, D., Wetterhall, F., Ireson, A. M., Chandler, R. E., Kendon, E. J., Widmann, M., Brienen, S., Rust, H. W., Sauter, T., Themessl, M., Venema, V. K. C., Chun, K. P., Goodess, C. M., Jones, R. G., Onof, C., Vrac, M. & Thiele-Eich, I. 2010. Precipitation downscaling under climate change. Recent developments to bridge the gap between dynamical models and the end user. Reviews of Geophysics 48, RG3003.

Furrer, E. M. & Katz, R. W. 2008. Improving the simulation of extreme precipitation events by stochastic weather generators. Water Resources Research 44, art. W12439.

---

## Referee Comment (RC3) · Anonymous Referee #3 · 16 Jul 2016

This manuscript tests the sensitivity of an analogue downscaling method for precipitation to an extension of the potential analogue situations to 6-, 12- and 18-hour shifted analogue dates, with a high temporal resolution archive. Such an extension appears to increase the skill of the method as applied to 6 stations in Switzerland. The manuscript thus addresses a relevant scientific question within the scope of HESS. As far as I know, this idea have not been explored before and the conclusions make it sound appealing. The method seems sound, even if not clearly enough detailed at times, and the results support well the conclusions. The title wrongly suggests that these conclusions are valid only in a forecasting context, while they actually have much more general implications. A proper literature review is missing, and the number of tables is

too large. These conclusions are detailed in sections below. I therefore recommend the manuscript to be reconsidered after major revisions.

**General comments**

1. Structure of the paper: the manuscript is strangely structured and should undergo many changes to improve its readability:

   - The introduction is for a large part a presentation of the analogue downscaling tool used and thus belongs to the "methods" section,

   - One consequence is a lack of a proper introduction, with a scientific context, a proper literature review, and so on. Positioning this study within the wider issue of the archive length (as noted by Referee #1) – that has been studied for quite some time (see e.g. Ruosteenoja, 1988; Van den Dool, 1994), and notably recently by Radanovics et al. (2013) – would be quite relevant. More generally, various properties of the analogue approach have been largely studied recently, like temporal transferability (see e.g. Dayon et al., 2015; Caillouet et al., 2016), or spatial transferability (Chardon et al., 2014), in more climate-oriented contexts. The moving time window proposed here would be perfectly suited for improving the different variants of the analogue methods in such contexts, and this should also appear in the revised version of the introduction or conclusion.

   - There is no discussion section, while several paragraphs from the "data and methods" section – or even the introduction – should belong to such a discussion section,

   - Several paragraphs from the "Results" section – or from section 4 – should belong to either the "methods" section or the "discussion" section.
2. Notations: Please use conventional abbreviations for commonly used quantities: Teweles-Wobus Score → TWS or S1, continuous ranked probability (skill) score → CRP(S)S, root mean square error → RMSE

3. Number of tables: there are much too many tables in the manuscript that could be either be put in a supplementary material or summarized through graphs.

4. Tense: please use the present tense for all description and analysis of the work carried out.

**Specific comments**

1. L3, "on the geopotential [...] gradients": please rephrase

2. L7-8: sentence without verb

3. L15-17: Is it not rather because heavy precipitation events are rarer?

4. L24-25: I don't understand

5. L33: Horton et al. (2016) is not published, even in GMD. You cannot therefore refer to developments and findings made in such a manuscript.

6. L33: Ben Daoud et al. (2016)

7. L34: What improvements? Please include this in the literature review.

8. L43-48: I presume this is for a 6h-6h precipitation totals, but it should be clearly stated here

9. L49-43: The archive should be described here.

10. L63: What are the parameters to calibrate? Please list them.

11. L70: Please either provide a peer-reviewed reference for this decomposition or detail it here.

12. L76: Please detail the computation of the climatological distribution.

13. L81-82, "otherwise [. . .] them". Please rephrase.

14. L92-97: Some parts of this paragraph belongs to the discussion section.

15. L93-94: I don't understand.

16. L111-112: Please remove the sentence or remove the corresponding par in the next section.

17. L113-114: Please justify the use of such an outdated global reanalysis (I understand this is partly for having a long time coverage). And add also the potential of using more recent and products with higher quality to the discussion.

18. L120, "validation"? Please describe in detail the experiment set-up.

19. L123, "based on [. . .] shown)": please detail or remove.

20. L131: again, Horton et al. (2016) is not available, so you should provide a description of the calibration procedure.

21. L133-136: This belongs to the results section.

22. L146-147: An increase with what?

23. L148-149, "the latter [. . .] selection": I don't understand.

24. L156-157, "This [. . .] distribution": already written above.

25. L162-167: This belongs to the discussion.

26. L168-174: This analysis is done for different classes of precipitation values. Whether this relates to the intensity of circulation dynamics is another issue.

27. L180-183: This belongs to the discussion.

28. L192-193: Figure 8 is not necessary. Please remove of put it in a supplementary material.

29. L202-206: This belongs to the discussion.

30. L208-214: This belongs to the methods.

31. L215, "performance scores": specify that these are CRPS values.

32. L219-220, "No relationship [. . .] criteria": I don't understand.

33. L246-250: Is it shown somewhere in the manuscript?

34. L273-281: This belongs to the methods section.

35. L286-298: This belongs to the methods section.

36. L350-354: Given recent studies on RCM biases, I have serious doubts that RCM precipitation is reliable enough for it to be use as observed series in this context.

37. Figure 11: The bottom right panel is identical to the one to its left. Plus, choosing a display set-up with rows showing accuracy and sharpness, respectively, would allow different y scales and increase the readability.

38. Table 13: The choice for preselecting these 4 points should be somehow justified.

**Technical corrections**

1. L2: "precipitation"

2. L145: "shape"

3. L162: "dynamics"

4. L209: "obligatory" → necessarily

**References**

Caillouet, L., Vidal, J.-P., Sauquet, E., and Graff, B.: Probabilistic precipitation and temperature downscaling of the Twentieth Century Reanalysis over France, Climate of the Past, 12, 635-662, doi:10.5194/cp-12-635-2016, 2016

Chardon, J., Hingray, B., Favre, A.-C., Autin, P., Gailhard, J., Zin, I., and Obled, C.: Spatial similarity and transferability of analog dates for precipitation downscaling over France, Journal of Climate, 27, 5056-5074, doi:10.1175/JCLI-D-13-00464.1, 2014.

Dayon, G., Boé, J., and Martin, E.: Transferability in the future climate of a statistical downscaling method for precipitation in France, Journal of Geophysical Research, 120, 1023-1043, doi:10.1002/2014JD022236, 2015.

Radanovics, S., Vidal, J.-P., Sauquet, E., Ben Daoud, A. Bontron, G.: Optimising predictor domains for spatially coherent precipitation downscaling, Hydrology and Earth System Sciences, 17, 4189-4208, doi:10.5194/hess-17-4189-2013, 2013.

Ruosteenoja, K.: Factors affecting the occurrence and lifetime of 500 mb height analogues: a study based on a large amount of data, Monthly Weather Review, 116, 368-376. doi:10.1175/1520-0493(1988)116<0368:FATOAL>2.0.CO;2, 1988.

Van den Dool, H. M.: Searching for analogues, how long must we wait?, Tellus A, 46, 314-324. doi:10.1034/j.1600-0870.1994.t01-2-00006.x, 1994.

---

## Editor Comment (EC1) · J. Seibert (Editor) · 12 Sep 2016

While all three reviewers appreciate the value of the study they also express important concerns regarding the manuscript:

1) The structure needs to be improved. As it is now the intro includes a lot of the methods and in the section results&discussion there is a mix of methods, results and discussion. Please clearly seperate these parts

2) Overall the manuscript would benefit from a language check. As partly noted by the reviewers there are quite same language issues. On point is the use of the tenses, which is rather inconsitent. Please use past tense for describing your study and present

tense for established knowledge (previous studies) (note, this is opposite t the recommendation of one of the reviewers, but is the more common use of tenses in scientific papers)

3) Previous work needs to be better referred to.

4) Extreme events are challenging for the AM. This should be better discussed.

5) The importance for hydrological applications could be better clarified

Best regards, Jan Seibert

––––––––––––––––––––

---

## Author Response (AR1)

**Reply to comments of referee #1**

First of all, thank you for carefully reading the manuscript and for the constructive feedback.

Some referee comments are recalled in italics and followed by the authors' responses. Others are addressed here, but without being recalled. The technical corrections or rephrasing are not discussed here, but will be performed.

- Introduction: The structure will be changed and the description of the method moved into "Data and method". Additional references will be added, and different variations will be presented, instead of relying on another paper to cover the literature review. The introduction will thus be rewritten.

- *The issue discussed in the manuscript goes beyond this "fixed/moving" window issue. It more generally consider influence of the size of the data set (including the length of the archive) on the prediction skill. I am not aware of works that explored this issue but references to those, if any, and associated results have to be presented in the introduction.*
  The issue of the archive length will be better described, according to the relevant literature.

- *The final objective of the work is to increase the prediction skill of the method. Other means are possible for this. They should be also mentioned.*
  We will mention other means to increase the prediction skill in the introduction.

- MTW strategy vs longer archive: We agree that the MTW might not be as performant as a 4 times longer archive, for the reasons provided. We indeed observed a gain in performance similar to doubling the archive. Your point on the transient climate is also interesting. We will add a discussion on these aspects in the paper.

- Number of analogs: we totally agree with the referee that when the choice of the predictors or the parameters are improved, leading to a better prediction, the optimal number of analog situations thus decreases. However, when the length of the archive increases, the optimal number of analogs increases too, for a better performance, up to a certain threshold. This has been demonstrated in the PhD of G. Bontron (2004, p. 227), and we can also see it in Fig.7 of Hamill et al. (2006). This increase in the number of analog situations results objectively in better performance skills in our case. The MTW enriches the pool of available situations, even though they are not fully independent. We will improve the clarity of this analysis. It is true though that the choice of predictors can be improved and auxiliary predictors are missing. The chosen methods are not the most recent ones, but are benchmarks as they have been used by several studies as references, and the results are easier to interpret than more elaborated methods. A note on that will be added.

- *Explain both terms of the decomposition. Another well-known decomposition of the CRPSS is that of Herbasch. Please mention it also and clarify the advantage of that of Bontron. It is not clear at this stage.*
  We will remove the analysis of the sharpness and accuracy, as it brings unneeded complexity without being very instructive.

- *Why using a non conventional notation for the CRPS score. Please use the classical one (CRPSS) or justify*
  It is the journal convention for variables in equations ("Multi-letter variables should be avoided. Instead use single-letter variables with subscript (e.g. $E_{RMS}$ instead of RMSE, or $E_T$ instead of ET)."). It has been changed elsewhere.

- Resolution of the atmospheric reanalysis: it was shown by some studies that the resolution of the reanalysis dataset does not improve significantly the performance of the analogy of the atmospheric circulation. However, a full analysis of this aspect is out of the scope of this publication, and is a topic we are working on right now. Nevertheless, we can assume that it does not alter the fact that we can find better analog situations at different hours of the day.

- Seasonal stratification: yes, we consider a seasonal stratification. It is mentioned in l. 36-39. This should be more clear after the change in the paper structure.

- *You use the mean intensity of rainfall as a proxy of "dynamism of the atmospheric situation". Another proxy could be the intensity of the variations within the geopotential fields (e.g. mean gradient value). Why did you not explore this? It would better fit to your "dynamism" concept.*
  Yes, that could be a possibility. However, even though the link between the dynamism and the precipitation amount is not direct, the interest in analyzing it this way is that it highlights improvements we are directly interested in: a better prediction of high precipitation amounts. This will be rephrased to focus more on the precipitation thresholds rather than the dynamism of the circulation pattern.

- Sharpness and accuracy: We will remove the analysis of the sharpness and accuracy, as mentioned previously.

- Figure 11: this figure is complex and not very instructive. It will be removed along with the sharpness and accuracy analysis.

- Tables: we agree that there are many tables. We will remove some unanalyzed data, such as the spatial windows in Tables 3, 4, 7, and 8 in order to group the remaining information. Tables 10 and 11 will also be removed along with the sharpness and accuracy analysis.

- The other unmentioned detailed issues will be fixed

**Reply to comments of referee #2**

The authors would like to thank Referee 2 for his/her positive comments on the manuscript.

Some referee comments are recalled in italics and followed by the authors' responses. Others are addressed here, but without being recalled. The technical corrections or rephrasing are not discussed here, but will be performed.

- Sampling of extremes: The topic of extreme values within the analog method should be addressed in details, we agree on that and plan to work on it. However, it is out of the scope

of this paper. The proposed MTW improvement does not change the limitation of the maximum observed values in the archive, but it is not the topic of the present paper. We will however add a note on that issue in the paper.

- Introduction and applications in hydrology: The introduction will be rewritten, as explained to referee #1, and references will be added. Some applications to hydrology will also be cited.

- *Given the focus of the paper on the sub-daily time step, I suggest to refer more on the needs in urban hydrology.*
  The sub-daily time step is introduced in the search of analog situations, so at the predictors level. The predictand remains a 24-h precipitation total, which limits the application to urban hydrology.

- How the sample sizes N1 and N2 are determined and optimized: Indeed, this has not been detailed here. We will provide more insight on the calibration procedure.

- The other unmentioned issues will be fixed

**Reply to comments of referee #3**

We would like to thank Referee 3 for his detailed and relevant review.

Some referee comments are recalled in italics and followed by the authors' responses. Others are addressed here, but without being recalled. The technical corrections or rephrasing are not discussed here, but will be performed.

- Introduction: as replied to referee #1, the introduction will be rewritten, with an improved literature review and context description. The method description will be moved to the methods section.

- *The title wrongly suggests that these conclusions are valid only in a forecasting context, while they actually have much more general implications.*
  We agree that the application of the MTW is not limited to forecasting. The title will be changed to a more generic term.

- The structure will be improved, with a better separation of methods, results and discussion.

- *Notations: Please use conventional abbreviations for commonly used quantities: Teweles-Wobus Score → TWS or S1, continuous ranked probability (skill) score → CRP(S)S, root mean square error → RMSE*
  It is the journal convention for variables in equations ("Multi-letter variables should be avoided. Instead use single-letter variables with subscript (e.g. $E_{RMS}$ instead of RMSE, or $E_T$ instead of ET)."). It has been changed elsewhere.

- The number of tables will be decreased, as explained to referee #1: "We will remove some unanalyzed data, such as the spatial windows in Tables 3, 4, 7, and 8 in order to group the

remaining information. Tables 10 and 11 will also be removed along with the sharpness and accuracy analysis, which is not very informative."

- *L15-17: Is it not rather because heavy precipitation events are rarer?*
We cannot exclude this argument, and it might be a mix of both factors. We will add a note on that aspect.

- *L24-25: I don't understand*
This reports to section 4, but might not be necessary in the abstract as it brings some confusion.

- *L63: What are the parameters to calibrate? Please list them.*
More details will be provided on the calibration procedure.

- *L70: Please either provide a peer-reviewed reference for this decomposition or detail it here.*
We will remove the analysis of the CRPS decomposition, as it brings complexity without being very informative.

- *L76: Please detail the computation of the climatological distribution.*
A description will be added.

- *L93-94: I don't understand.*
This will be removed as it relies on partial analysis.

- *L113-114: Please justify the use of such an outdated global reanalysis (I understand this is partly for having a long time coverage). And add also the potential of using more recent and products with higher quality to the discussion.*
See answer to referee #1. These points will be addressed in the discussion.

- *L168-174: This analysis is done for different classes of precipitation values. Whether this relates to the intensity of circulation dynamics is another issue.*
We will reformulate this section.

- *L192-193: Figure 8 is not necessary. Please remove of put it in a supplementary material.*
The figure will be removed.

- *L219-220, "No relationship [: : :] criteria": I don't understand.*
This will be removed.

- *L246-250: Is it shown somewhere in the manuscript?*
No it is not shown, as the figure globally is very similar to Fig. 10.

- *L350-354: Given recent studies on RCM biases, I have serious doubts that RCM precipitation is reliable enough for it to be use as observed series in this context.*
We agree with the referee and will change this sentence.

- Figure 11: We will drop the analysis of the CRPS decomposition, as it brings complexity without being very informative.

- *Table 13: The choice for preselecting these 4 points should be somehow justified.*
  You are right. These are simply the points surrounding the catchment. It will be explained.

- The other unmentioned issues will be fixed

**Significant changes in the manuscript**

The manuscript was heavily reprocessed, which makes the changes tracking (next pages) a bit useless… The major points are:

- The whole structure changed, following the referees' advices. It should be much clearer now.
- The introduction has been re-written and include more references to other works.
- The decomposition in sharpness/accuracy has been removed, along with its figure.
- 8 tables were removed and some information were integrated into a new figure.
- The discussion has been extended to cover several questions asked by the referees.
- Globally, there is almost not a sentence from the original document that hasn't changed.
- The document has also been through English editing.

[revised manuscript text omitted]

---

## Referee Report (RR1)

Review comments for "The analogue method for precipitation prediction: finding better analogues situations at a sub-daily time step" by Horton et al.

Recommendation: Major revision

The authors introduced a moving time window (MTW) for the analogue method so that better analogues at a different hour can be found for precipitation prediction in contrast to the use of analogues at fixed hours of the day in standard analogue method. They found that the MTW with the shorter archive on a sub-daily time step improved the analogy criterion values across the entire distribution of analogue dates and the skill of precipitation prediction in comparison with the standard analogue method with longer archive on a daily time step. In particular, the improvement in prediction skill is greater for days with heavy precipitation. The topic is important and has great implications for operational precipitation forecasting and impact studies associated with the hydrological community. The only constraint is that the implementation of such method requires the availability of sub-daily time series, which may not always exist.

I have several major comments. First, some necessary information regarding the the presented analyses should be provided. For example, what season are the results shown in Figures 4, 5,6,7,8, 10 based on? The authors mentioned in Table 1 that the selection of analogue candidate is limited to the 4 months centered around the target date for every year. However, it is not clear what season the presented analyses focused on. Also, it seems to me that the entire assessment is performed in the prognosis context. The authors mentioned "prediction" several times throughout the paper. No matter for a 47-year archive (1961-2008) or reduced 25-year archive (1982-2007), it is not clear if the authors used part of the archive for calibration and part of the independent period for validation. If it is real "prediction", what period of data is the prediction performed on? All these details should be clearly described in the method section. Second, the paper, especially the results and discussion sections, is not well structured. These sections are divided into many small sub-sections. The content should be better organized and integrated to convey clear message. One example is, the discussion of Figure 4 and 5 appears in both section 3.1 and 3.3. Third, the text needs to be improved in terms of the logic, transition, grammar and wording. Some sentences are really long, confusing, and quite hard to understand (see some specific comments below).

Specific comments:

1. P1, line 6-7: confusing sentence, how about "the main reason for the use off daily precipitation time series is the length of their available archives, …
2. P1, Line 7-9: "However, it is … at a different time of day". Long and confusing sentence. should rephrase it.
3. P2, Line 22-23: "since they are based on observed situations with consistent spatial distribution" – consistent with what? Do you mean between target day and analogue dates? "as long as the analogue dates chosen for a region are the same" – same compared to what? When the target day changes, I think the analogue dates will change accordingly.

4. P3, Line 2: "even for much higher orders of magnitude" – do you mean even longer archive?
5. P3, Line 2-4: "Hopefully" – better to use "fortunately" based on context. Also, need reference for the statement "it appears that … 10° to 20°".
6. P3, Line 16-19: "Therefore, if the reduction of the archive … to an increase in performance". – very confusing sentence, please consider rephrasing it.
7. P3, line 25: "similar conclusions" – what is the conclusions? - in creasing the prediction skill?
8. P5, line 11-12: why MTW can not be applied to the 2$^{nd}$ level of analogy?
9. P.6, line 9 for Figure 2: why not just keep candidate 24-h precipitation fixed from 6h to 30h, but allowed to choose the analogues on 6h, 12h, 18h, 24h, 30 h for both Z500 and Z1000? That allows you to choose the analogues on multiple time steps but within the 24-h window consistent with conventional method. What is the purpose to have the varying 24h precipitation totals if the main objective is to find the better analogues to predict the same target day precipitation?
10. P6, Line13: confusing sentence "no constraint … in order to restrict."
11. P6, Line 27- 33, it is not clear how the method is implemented. The authors should provide a diagram to show the method. More details are preferred, such as do you just pick one best grid among four, what time lapse is allowed, how the temporal profile of best proxy is used to disaggregate? If you use the proxy variables from NCEP/NCAR reanalyses, why not directly use the precipitation from NCEP/NCAR reanalyses?
12. P7, line 13: Is the four points for geopotential height used to calculate the height gradient in both directions?
13. P8, Line 9: what does "globally significant" mean? Significant at what level?
14. P8, section 3.2.1: It is not clear to me how the distribution of the analogy criterion for different analogue ranks is constructed. So for any target day, if 50 analogue dates are selected (50 ranks in total), each analogue date should have only one S1 value based on their similarity in geopotential fields.
15. P9, line 6: "the number of candidate situations did not increase", but from table 3, N1 for 2Z-2MI is larger than N1 for 2Z.
16. P9, Line 10-11: could this because RMSE is not a good metric to assess the similarity for moisture fields?
17. P9, Line 13: why it is "prediction"? I think the entire assessment so far is in a prognosis context. Do you reselect the analogue dates for blue bar (MTW algorithm) in Fig. 4 and 5?
18. P9, Line 15: It will be good to test if the improvements of MTW and MTW-r over the static approach is significant?
19. P9, section 3.3.1: Fig.11 also indicates that the spread of difference of the CRPSS performance score is quite larger. It is not correct to say that the performance score was improved for days with high precipitation. The statement should be based on the average performance. Again, for Fig.11, it is not clear to me what each point represents. Do the points represent the analogue dates with precipitation amount in the specific categories? Then the total number of pints in figures are equal to the total number of analogues selected?

20. P9, section 3.3 and 3.3.1: when author say "prediction skill", does the author mean the use of calibrated parameters for independent data set?
21. Same as #10, it would be good to show a map about the method 2 to help the reader understand what is concluded in section 3.4 and table 5.

---

## Referee Report (RR2)

Review of manuscript :
**The analogue model for precipitation forecasting : finding better analog situations at a sub-daily time step.**
**By Horton, P. and coll.**

General comment
The manuscript has been restructured, simplified and largely improved. The manuscript is well organized, and the interest of the MTW approach clearly highlighted and discussed. Some points have for me still to be clarified/precised. See minor comments below.
With these clarifications/corrections, the manuscript is for me of the quality required to be accepted for publication in HESS.

Detailed comments
P. 3 ln 3. You mention "AMs can also be combined with other methods (e.g. Chardon et al., 2014)." This I not what I have retained from the paper you mention. You perhaps refer to the following manuscript, currently in review in HESS : Chardon, J., Hingray, B., and Favre, A.-C.: An adaptive two-stage analog/regression model for probabilistic prediction of local precipitation in France, Hydrol. Earth Syst. Sci. Discuss., doi:10.5194/hess-2017-62, in review, 2017.

This comment applies also for p10. Ln 23 - Another possible approach is to combine AMs with other methods (e.g. Chardon et al., 2014). > This is likely to be not the good ref. to be mentioned there

P8 – first paragraph : there are some repetitions > please reduce / simplify

P9 ln9 : "*The prediction skill for the CP was almost always improved further by reducing the time step of the MTW, but not of the same magnitude*" > do you compare here the results obtained for the two different reanalyses (MERRA / ERA20C) or for different MTW windows (for a given reanalysis) ?

P9 ln22 : you mention "After the introduction of the MTW, the performance score was generally further improved with reduced CRPS for days with higher precipitation than for non-rainy days and small precipitation values" > this results seems to be expected as the CRPS is expected to have greater values when the precipitation amount to be predicted is higher (even if the relative sharpness of the prediction (which roughly corresponds the standard deviation of the distribution divided by its mean) is the same). A comment would be welcome here and likely also in section 4.1.

p10 – ln 30 > rephrase (not clear) : *These higher numbers of analogues were objectively chosen by using the calibration procedure (Sect. 2.3) in order to increase the prediction skill of the methods.*

P11. Ln 23 > rephrase (not clear) : *With the introduction of the MTW, the performance loss related to an eventual reduction of the archive length to meet the length of the sub-daily precipitation archive was indeed compensated.*

P11 ln 28 :: clarify what you mean with : "*Moreover, rather strong serial correlations between successive sub-daily circulation patterns are expected*". I guess you want to say that 2 consecutive 3hourly time steps present in some way redundant information.

P12 – ln 3>7 (last § of section 4.3) *"One can question the interest of using moving daily totals when, for example, 6-h precipitation series can be predicted instead"* I do not see the interest of this paragraph. For me, this is out of scope of the work and could be removed. You focus on the prediction of daily totals, not on subdaily ones. If this paragraph has to be kept, clarify what you mean with : "However, the 6-h time series generated by the AM might not accurately represent the intra-daily precipitation distribution"or variability » > do you refer to the difficulty to produce relevant "multiple 6h00" sequences (e;g. daily sequences with relevant temporal subdaily profiles) ? If yes, this question applies also for predictions produced at a daily time step (i.e. what is the temporal relevance of sequences of 3 days when generated with a daily model?) Could you please clarify this point ?

I do not understand also the last sentence of this paragraph. *"Finally, when using a reconstructed precipitation archive, the errors in intra-daily precipitation distributions have a smaller impact on 24-h totals than on 6-h totals."* Please Clarify

P12. L10 : please clarify what is the time period you consider for daily data in this section.

P12. L 11 : I suggest to change the end of the paragraph "*Therefore, the idea is to reconstruct longer archives of …* "for "One possible approach to get such long time series is to reconstruct moving 24-h totals from existing standard daily precipitation series. For this purpose, disaggregation techniques can be used. In this study, we consider the interest of such reconstruction approach using in turn two simple disaggregation methods".

P12. Ln 23 : I do not understand this sentence : Please rephrase / clarify : "*Time lapses from -12 h to +12 h between both series were introduced to consider the significant distance separating the weather stations and the reanalysis grid point.*

P12. Ln 24 : "*The best proxy variable, precipitable water, was identified through correlation analyses on non-zero values with the 6-h precipitation time series* »
On which period did you do the correlation analysis ?
To which variable refer the "non-zero values" (precipitation ? moisture ?) ?

P12. Ln 27 "*A slight improvement was obtained for the second method*" > Do you mean improvement from the constant "disaggregation" method ? or from the smaller period archive configuration ? Please clarify.

P13 – ln 27 : the logic of this paragraph and of the next one is not optimal.
For me, the critical issue you want to highlight here is the size of the pool of analogs candidate which has to be the largest as possible. Hence, MTW can increase this size (inflation). Another possibility relies on long archives of daily precipitation but requires estimates of sub-daily structures. > Here you can introduce the issue of the quality of the chronology of precipitation at a high resolution (e. 3hourly) time step.

As a perspective, you could also say that another (simpler) strategy would be to use a database relying on two different data sources :
- MTW for the period with 3hourly data (30yrs * 8 equivalent data amount)
- Classical fixed window approach for the period with only daily data (1900> 1980 = 80 years of additional daily data)

P13 – ln 32 : I do not really understand what you suggest as a perspective in the following paragraph.

*"The precipitation data archives of high temporal resolution have increased over time. Other possible sources of such archives is the establishment of precipitation reanalysis at a regional scale in addition to the use of reanalysis-driven regional climate models or limited area models over a long period. Even though outputs from these models might be biased or not accurate enough, information regarding the timing of the precipitation events could be useful in disaggregating the station time series."*

> I understand the potential interest of precipitation estimates from reanalysis-driven climate models over a long period. But, what do you mean with *"establishment of precipitation reanalysis at a regional scale"* and what is the difference / interest when compared to *"reanalysis-driven regional climate models or limited area models over a long period"*. (why do you use the term "in addition") – what is typically the long period you have in mind ? Do you suggest to use these simulated precipitation data as a proxy to disaggregate daily precipitation observations ? Please Clarify.

P14 – ln 4 > please clarify this paragraph.
I do not understand your statement : *"this [selection] improvement has the potential for application to long meteorological archives."*
Is the main idea to say that the MTW can be used to have a better diagnosis of the current / future weather situation even if no high resolution and high quality precipitation data are available ?
Is the "long archive" issue a key issue here ?
Could you explain why we do not need such quantitative values of precipitation for these analog dates ? (do you consider that you may have other observed values/events in some other historical database (flood events, other hydrometeorological proxies) that allow you to inform on the likely severity of the current weather situation to predict ?)

You finally mention : "Finally, some other predictands might not need sub-daily total values but point observations such as hail or extreme wind gusts, which make them easier to use with the MTW." Do you suggest that the "better easiness" relies on the fact that each point observation can be attributed without any disaggregation issue to each of the different MTW window of a given day ? Please clarify or rephrase…

---

## Author Response (AR2)

**Reply to referee 1: Review of "The Analogue Method for Precipitation Prediction: Finding Better Analogue Situations at a Sub-Daily Time Step." (HESS-2016-246) by Horton et al. 2016.**

This is my first time reviewing this manuscript, though it is my understanding that it was previously reviewed by others, with major revisions requested. I have not studied the comments of other reviewers, with the desire to provide guidance as independent as possible.

My recommendation, with regret, is rejection. The article is needs significant reorganization and rethought, above what is required for a major revision. Some of my most significant issues include: (a) it provided detail of a minor improvement to a rather antiquated post-processing methodology; (b) it uses data sets for which there are better alternatives; (c) it doesn't describe all of the procedures clearly; (d) it doesn't consider other alternatives as controls against which to evaluate the methodology. In sum, I think the authors need to re-evaluate their research from top to bottom.
All the analyses were performed again with a more recent dataset. It allowed us to change the structure of the study and then to restructure the paper to gain in clarity. The procedures were then better explained. The other mentioned issues are discussed below.

Here are some more details on my most significant concerns.

Poor organization. Journal articles are generally minor variations on a standard organization, with an introduction, data, methods, results, conclusion. Here are some of the issues with such sections:
We improved the structure and simplified the overall logic of the study.

Introduction. There is now a rather rich body of literature on the statistical post-processing to produce probabilistic precipitation forecasts. These include Gamma-distribution fitting methods, Bayesian Model Averaging, Extended Logistic Regression, and more. With the use of an older analog approach, one wonders why such an approach is considered, in the first instance, and how one should place in context the results to follow. Without a thorough review of other possible alternatives and some explanation of why your approach is being considered despite these others, the reader is left wondering why they should bother with continuing to read the rest of the manuscript.
The present use of the analogue method is not a statistical post-processing of the precipitation. It is a statistical adaptation technique, which is classified as a downscaling approach (which can be a language abuse), as it provides a statistical precipitation prediction (or ev. forecast) based on large scale predictors describing the synoptic scale conditions. The precipitation output from the NWP model (or GCM, RCM, … according to the context) is not even considered in the method, as it would be in statistical post-processing methods. The goal of using an analogue method is to predict local precipitation based on the outputs of a global NWP model, without the use of a limited-area model. It brings an alternative approach that can complement the forecast provided by limited-area models in operational forecasting e.g., or the RCMs in the context of climate change impact studies.
There was a European project (COST VALUE, http://www.value-cost.eu/) that aimed at comparing different downscaling methods. The results of the project are not published yet, but show that there is not an overall best downscaling technique, as the different techniques have some advantages and weaknesses in different characteristics. The analogue method was found to be strong in some characteristics and weaker in others. However, such comparison is out of the scope of the present study.

As we show in the introduction, there are several articles published in the very last years using the analogue method, and we are aware of its use in different ongoing studies. Thus, the goal of the paper is more to provide an improvement to the users of the method. We refer to the paper of Maraun et al (2010) for alternative downscaling methods.

Data. What information was being used for the forecast was not made clear, and this is crucial information to find out right away. See (2a) below for more.
We added this information. By the way, the study takes place in a perfect prog context, not a forecasting context.

Methods. Section 2.2 and 2.6 seem to be describing two different methods. There should be one, single, clear, unambiguous description of the methodology to be used.
Section 2.2 describes the analogue method itself, and section 2.6 (now 2.3) how the parameters of the method are calibrated. We tried to make it more clear and restructured section 2.
Also in this section, while verification using CRPSS is relatively standard, so are other verification diagnostics like reliability diagrams, which are not presented.
Reliability diagrams were added (Figure 12).

Choice of data sets.

Forecast data. What is used as the forecast, and why, is not clear. Is another analysis from the NCEP-NCAR reanalysis time series used as a surrogate for a numerical forecast? Is ECMWF, or other model forecast data used? I could not tell. If NCEP-NCAR reanalysis data is used, this then begs the question: why? This would not be a practical forecast methodology, where one needs information in advance of the event. If a numerical weather prediction forecast is used, then there are potential issues of forecast bias; the forecast model and the analysis may be different in character, leading to the issue of whether perfect-prog (your analog) type approaches are suitable or whether more model-output statistics approaches are needed.
This study takes place in the perfect prog context. There is no forecast here, only prediction on an independent period (the validation period) using the reanalysis data. Then, when applied in real-time forecasting, NWP forecasts are considered. The issue of the biases are known and for this reason: (1) the model used for the NWP forecast and the one used for the reanalysis product should be as similar as possible, and (2) the main predictors are geopotential heights, which are robust and not too much model-dependent (this is more an issue for moisture variables). Moreover, the geopotential heights are compared in terms of their shape and not absolute value. Some MOS use of AMs also exist, provided that a long enough archive is available. In that context, the MTW could also be used and provide improvements. When applied in a climate context, AMs rely on GCM or ev. RCMs outputs. In this study, we stick to the perfect prog context and show an improvement that can then be applied to the forecasting context or to climate impact studies.

Reanalysis data. There are more modern, more accurate reanalysis data sets available now such as ERA-20C, available at higher temporal resolution (3 hourly), which seems to be crucial for an article examining the usefulness of temporal shifts of the data. You dismiss this in your section 4.6 with an older reference, but I think given the focus in this article on temporal shifts, you need to reconsider higher temporal resolution reanalysis data.
You are right that it is important considering a 3 hourly dataset. We have done all analyzes again using ERA-20C and MERRA-2 reanalysis datasets. Doing so allowed us to consider another workflow and to simplify the analyses and then the paper.

Observation data. While the geographic details of the observation locations are definitely different from many other locations in Europe, there are still other locations in the mountains. Why not consider approaches that supplement the training data with other locations' observations, potentially allowing you to get more without needing a very lengthy reanalysis? You may have objections to this, but at the least it would be worth explaining your choices.

We are not sure to get your point. The AM exploit the intrinsic link between the synoptic-scale situation and the local precipitation. This relationship is very location-specific, as the meteorological influences will not be the same when considering a mountainous environment in another country. The location and characteristics of the driving elements, such as the low pressure centers and the fronts, will not be the same, neither the characteristics of the precipitation climatology. We can therefore not exploit remote data.

Methodology. Finding some other reference methodology, e.g., Bayesian Model Averaging, would certainly be desirable. Another logical control would be unadjusted ensemble forecasts from the ECMWF ensemble prediction system. This would allow you to have a point of comparison against which to judge the analog methodology.

As stated previously, the study does not take place in a forecast context, and thus cannot be compared to ensemble forecasts provided by NWP models.

But even considering the analog methodology in isolation, one wonders why you chose the particular approach to selecting analogs. In particular, I found myself wondering why, for example, you didn't use canonical correlation analysis approaches to determine what information in the reanalysis data set was most directly related to precipitation variability. Analog approaches, in my experience, are not very "efficient" with their training data; either a previous day's data is selected as an analog date, or not. This, then, drops on the floor all the other data, which may yet have some useful information. To use an analog approach, then, it seems especially incumbent to have demonstrated that you have chosen the most important predictors that will be used in selecting past dates.

Your version of the analogue method seems different to what we use here, so it is not easy to answer specifically to your points. However, here are some response elements:

- The choice of predictors is based on previous work (given as references) that made an intensive comparison of many variables. Moreover, the relevance of the geopotential heights and the moisture variables was shown by Horton (2012) to be the most relevant variables for the region of interest.
- A canonical correlation analysis cannot be performed here, as the geopotential heights are analyzed in terms of shape (Teweless-Wobus criteria) over a spatial window rather than absolute values.
- In the AM considered, the selection of analogue dates are not allowed within the same year as the target date. Thus, a selection of the previous date as analogue is not allowed.

In summary, I regret recommending rejection. I hope the authors will constructively use this feedback in the spirit intended.

**Reply to referee 2: Review comments for "The analogue method for precipitation prediction: finding better analogues situations at a sub-daily time step" by Horton et al.**

Recommendation: Major revision

The authors introduced a moving time window (MTW) for the analogue method so that better analogues at a different hour can be found for precipitation prediction in contrast to the use of analogues at fixed hours of the day in standard analogue method. They found that the MTW with the shorter archive on a sub-daily time step improved the analogy criterion values across the entire distribution of analogue dates and the skill of precipitation prediction in comparison with the standard analogue method with longer archive on a daily time step. In particular, the improvement in prediction skill is greater for days with heavy precipitation. The topic is important and has great implications for operational precipitation forecasting and impact studies associated with the hydrological community. The only constraint is that the implementation of such method requires the availability of sub-daily time series, which may not always exist.

I have several major comments. First, some necessary information regarding the the presented analyses should be provided. For example, what season are the results shown in Figures 4, 5,6,7,8, 10 based on?

We added a description of the calibration and validation periods. The results are for all seasons over the calibration or validation periods, covering several years.

The authors mentioned in Table 1 that the selection of analogue candidate is limited to the 4 months centered around the target date for every year. However, it is not clear what season the presented analyses focused on.

As explained above, the analyses are for all seasons over several years. It is true that the search for analogues is restricted to the 4 months centered around the target date, but the target dates cover all days over several years.

Also, it seems to me that the entire assessment is performed in the prognosis context. The authors mentioned "prediction" several times throughout the paper. No matter for a 47-year archive (1961-2008) or reduced 25-year archive (1982-2007), it is not clear if the authors used part of the archive for calibration and part of the independent period for validation. If it is real "prediction", what period of data is the prediction performed on? All these details should be clearly described in the method section.

We now clarified the use of independent data in the section "2.3 Calibration of the analogue method". It is indeed a prediction over a validation period, but still in the perfect prognosis framework, rather than a forecast.

Second, the paper, especially the results and discussion sections, is not well structured. These sections are divided into many small sub-sections. The content should be better organized and integrated to convey clear message. One example is, the discussion of Figure 4 and 5 appears in both section 3.1 and 3.3.

All the analyses were performed again with a more recent dataset. It allowed us to change the workflow of the study and then to restructure the paper to gain in clarity. Some results of secondary importance were moved to the discussion.

Third, the text needs to be improved in terms of the logic, transition, grammar and wording. Some sentences are really long, confusing, and quite hard to understand (see some specific comments below).

We tried to improve the language and better explain some points. The paper was corrected by Elsevier's English editing services.

Specific comments:

1. P1, line 6-7: confusing sentence, how about "the main reason for the use off daily precipitation time series is the length of their available archives, …
   This was changed.

2. P1, Line 7-9: "However, it is … at a different time of day". Long and confusing sentence. should rephrase it.
   This was changed.

3. P2, Line 22-23: "since they are based on observed situations with consistent spatial distribution" – consistent with what? Do you mean between target day and analogue dates? "as long as the analogue dates chosen for a region are the same" – same compared to what? When the target day changes, I think the analogue dates will change accordingly.
   This sentence was removed as it is a bit out of context.

4. P3, Line 2: "even for much higher orders of magnitude" – do you mean even longer archive?
   This was changed.

5. P3, Line 2-4: "Hopefully" – better to use "fortunately" based on context. Also, need reference for the statement "it appears that … 10° to 20°".
   This was changed and references added.

6. P3, Line 16-19: "Therefore, if the reduction of the archive … to an increase in performance". – very confusing sentence, please consider rephrasing it.
   This was removed.

7. P3, line 25: "similar conclusions" – what is the conclusions? - in creasing the prediction skill?
   This was removed.

8. P5, line 11-12: why MTW can not be applied to the 2nd level of analogy?
   Yes it can be applied to the 2nd level of analogy. Thus, this comment was removed.

9. P.6, line 9 for Figure 2: why not just keep candidate 24-h precipitation fixed from 6h to 30h, but allowed to choose the analogues on 6h, 12h, 18h, 24h, 30 h for both Z500 and Z1000? That allows you to choose the analogues on multiple time steps but within the 24-h window consistent with conventional method. What is the purpose to have the varying 24h precipitation totals if the main objective is to find the better analogues to predict the same target day precipitation?
   This section was rephrased for clarity.

10. P6, Line13: confusing sentence "no constraint … in order to restrict."
    This was removed.

11. P6, Line 27- 33, it is not clear how the method is implemented. The authors should provide a diagram to show the method. More details are preferred, such as do you just pick one best grid among four, what time lapse is allowed, how the temporal profile of best proxy is used to disaggregate? If you use the proxy variables from NCEP/NCAR reanalyses, why not directly use the precipitation from NCEP/NCAR reanalyses?
    The importance of this analysis was decreased and it moved to the discussion (Sect. 4.4), without providing all the details. It is indeed not the main message of the paper and brought some confusion. Some additional details on the time lapse and the non-consideration of precipitation were however added.

12. P7, line 13: Is the four points for geopotential height used to calculate the height gradient in both directions?

Yes. It has been clarified.

13. P8, Line 9: what does "globally significant" mean? Significant at what level?

This is not present anymore in the paper.

14. P8, section 3.2.1: It is not clear to me how the distribution of the analogy criterion for different analogue ranks is constructed. So for any target day, if 50 analogue dates are selected (50 ranks in total), each analogue date should have only one S1 value based on their similarity in geopotential fields.

Yes, each analogue date has a unique S1 value. These distributions for the different analogue ranks are obtained when applying the method on a long period. Then, for a considered analogue rank, we have multiple values of S1 corresponding to different target dates. We tried to clarify in the manuscript.

15. P9, line 6: "the number of candidate situations did not increase", but from table 3, N1 for 2Z-2MI is larger than N1 for 2Z.

Yes, but here it was with the original parameters, so without an increase in N1. It has been clarified.

16. P9, Line 10-11: could this because RMSE is not a good metric to assess the similarity for moisture fields?

No, it is because the use of an MTW does not increase the sample size in this case, the second level of analogy only subsamples in the dates provided by the first level. A comment was added.

17. P9, Line 13: why it is "prediction"? I think the entire assessment so far is in a prognosis context. Do you reselect the analogue dates for blue bar (MTW algorithm) in Fig. 4 and 5?

We use the term "prediction" instead of "forecast" because it is not operational forecast, but we stay in the perfect prog context. Predictions were established for a validation period with independent data. Fig. 4 and 5 do not exist anymore and were replaced by Fig. 9 and 10 that are different.

18. P9, Line 15: It will be good to test if the improvements of MTW and MTW-r over the static approach is significant?

The approach changed and the calculation are now done on a calibration period and an independent validation period in order to validate the potential gains.

19. P9, section 3.3.1: Fig.11 also indicates that the spread of difference of the CRPSS performance score is quite larger. It is not correct to say that the performance score was improved for days with high precipitation. The statement should be based on the average performance. Again, for Fig.11, it is not clear to me what each point represents. Do the points represent the analogue dates with precipitation amount in the specific categories? Then the total number of pints in figures are equal to the total number of analogues selected?

We improved the description of the analysis. It is indeed a difference on the scores between the conventional approach and the MTW. All points correspond to target dates over the whole period. It means e.g. that over the whole period, the prediction performance of most days with a target value above 30mm/d was improved. The mean is represented by the star and then shows an average improvement.

20. P9, section 3.3 and 3.3.1: when author say "prediction skill", does the author mean the use of calibrated parameters for independent data set?

Now we do use an independent dataset (the validation period, VP).

21.     Same as #10, it would be good to show a map about the method 2 to help the reader understand what is concluded in section 3.4 and table 5.

As we wrote above, these results were removed to focus on the main results of the study.

**List of relevant changes**

- There is now a distinct calibration period and an independent validation period.
- All the analyses were performed again with more recent datasets: MERRA-2 and ERA-20C, which allowed a 3-h time step.
- The influence of the MTW time step (3, 6, 12 h) is analyzed throughout the paper.
- Reliability diagrams were added.
- The structure of the study changed, which resulted in a better organized paper gaining in clarity.
- The analyses of secondary importance were removed to focus more on the main results of the study.
- All the text has gone through significant editing, as one can see in the marked manuscript.

[revised manuscript text omitted]
. −12 h −6 h 0 h +6 +12 h 1 0.668 0.669 0.684 0.683 0.670 2 0.669 0.669 0.683 0.681 0.669 3 0.662 0.673 0.691 0.682 0.673 4 0.666 0.671 0.688 0.681 0.668 1 0.672 0.673 0.684 0.684 0.675 2 0.674 0.674 0.683 0.682 0.672 3 0.662 0.673 0.691 0.682 0.673 4 0.666 0.671 0.689 0.681 0.668 1 0.675 0.675 0.679 0.678 0.671 2 0.681 0.690 0.691 0.677 0.664 3 0.665 0.680 0.693 0.683 0.675 4 0.675 0.694 0.706 0.681 0.659 1 0.688 0.687 0.667 0.655 0.652 2 0.697 0.699 0.669 0.644 0.644 3 0.686 0.708 0.689 0.655 0.648 4 0.696 **0.721** 0.696 0.643 0.636~~

---

## Author Response (AR3)

**Reply to referee 1**

**General comment**

The manuscript has been restructured, simplified and largely improved. The manuscript is well organized, and the interest of the MTW approach clearly highlighted and discussed. Some points have for me still to be clarified/precised. See minor comments below.

With these clarifications/corrections, the manuscript is for me of the quality required to be accepted for publication in HESS.

Thank you for your positive feedback.

**Detailed comments**

P. 3 ln 3. You mention "AMs can also be combined with other methods (e.g. Chardon et al., 2014)." This I not what I have retained from the paper you mention. You perhaps refer to the following manuscript, currently in review in HESS : Chardon, J., Hingray, B., and Favre, A.-C.: An adaptive two-stage analog/regression model for probabilistic prediction of local precipitation in France, Hydrol. Earth Syst. Sci. Discuss., doi:10.5194/hess-2017-62, in review, 2017.

Yes, you are right. We actually wanted to refer to his PhD thesis, but it is anyway better to refer to this new paper in discussion.

This comment applies also for p10. Ln 23 - Another possible approach is to combine AMs with other methods (e.g. Chardon et al., 2014). > This is likely to be not the good ref. to be mentioned there

Yes, thank you for mentioning.

P8 – first paragraph : there are some repetitions > please reduce / simplify

The paragraph has been simplified.

P9 ln9 : "The prediction skill for the CP was almost always improved further by reducing the time step of the MTW, but not of the same magnitude" > do you compare here the results obtained for the two different reanalyses (MERRA / ERA20C) or for different MTW windows (for a given reanalysis) ?

It is for different MTW time steps. This has been specified.

P9 ln22 : you mention "After the introduction of the MTW, the performance score was generally further improved with reduced CRPS for days with higher precipitation than for non-rainy days and small precipitation values" > this results seems to be expected as the CRPS is expected to have greater values when the precipitation amount to be predicted is higher (even if the relative sharpness of the prediction (which roughly corresponds the standard deviation of the distribution divided by its mean) is the same). A comment would be welcome here and likely also in section 4.1.

Based on your suggestion, the following sentence was added: "This can be expected, as the CRPS values are higher when the precipitation amount to be predicted is higher."

A comment was also added in section 4.1: "were improved to a greater extent for days with heavier precipitation (which are related to higher CRPS values)"

p10 – ln 30 > rephrase (not clear) : These higher numbers of analogues were objectively chosen by using the calibration procedure (Sect. 2.3) in order to increase the prediction skill of the methods.

This sentence has been removed, as it is implicit to the fact that we consider the "optimal numbers of analogues"

P11. Ln 23 > rephrase (not clear) : With the introduction of the MTW, the performance loss related to an eventual reduction of the archive length to meet the length of the sub-daily precipitation archive was indeed compensated.

This sentence has been removed as it doesn't bring much information

P11 ln 28 :: clarify what you mean with : "Moreover, rather strong serial correlations between successive sub-daily circulation patterns are expected". I guess you want to say that 2 consecutive 3hourly time steps present in some way redundant information.

This has been rephrased to : "Moreover, consecutive intra-daily situations are expected to be correlated and thus to present redundant information."

P12 – ln 3>7 (last § of section 4.3) "One can question the interest of using moving daily totals when, for example, 6-h precipitation series can be predicted instead" I do not see the interest of this paragraph. For me, this is out of scope of the work and could be removed. You focus on the prediction of daily totals, not on subdaily ones. If this paragraph has to be kept, clarify what you mean with : "However, the 6-h time series generated by the AM might not accurately represent the intra-daily precipitation distribution"or variability » > do you refer to the difficulty to produce relevant "multiple 6h00" sequences (e;g. daily sequences with relevant temporal subdaily profiles) ? If yes, this question applies also for predictions produced at a daily time step (i.e. what is the temporal relevance of sequences of 3 days when generated with a daily model?) Could you please clarify this point ? I do not understand also the last sentence of this paragraph. "Finally, when using a reconstructed precipitation archive, the errors in intra-daily precipitation distributions have a smaller impact on 24-h totals than on 6-h totals." Please Clarify

This paragraph has been removed.

P12. L10 : please clarify what is the time period you consider for daily data in this section.

This has been added: "06:00 h UTC to 06:00 h UTC the following day"

P12. L 11 : I suggest to change the end of the paragraph "Therefore, the idea is to reconstruct longer archives of … "for "One possible approach to get such long time series is to reconstruct moving 24-h totals from existing standard daily precipitation series. For this purpose, disaggregation techniques can be used. In this study, we consider the interest of such reconstruction approach using in turn two simple disaggregation methods".

Thanks, we changed the end of the paragraph.

P12. Ln 23 : I do not understand this sentence : Please rephrase / clarify : "Time lapses from - 12 h to +12 h between both series were introduced to consider the significant distance separating the weather stations and the reanalysis grid point.

This sentence has been removed and replaced by "and at different times of the day" in the following sentence: "Precipitable water and relative humidity at 1000 hPa, 925 hPa, or 850 hPa were assessed at the four points surrounding the catchment and at different times of the day"

P12. Ln 24 : "The best proxy variable, precipitable water, was identified through correlation analyses on non-zero values with the 6-h precipitation time series »

On which period did you do the correlation analysis ?

On the period 1982—2007. This has been added.

To which variable refer the "non-zero values" (precipitation ? moisture ?) ?

On non-zero precipitation values. This has been specified

P12. Ln 27 "A slight improvement was obtained for the second method" > Do you mean improvement from the constant "disaggregation" method ? or from the smaller period archive configuration ? Please clarify.

This has been rephrased to "The second method which relied on a proxy variable performed slightly better than the proportional distribution method, but not enough to obtain a relevant time series."

P13 – ln 27 : the logic of this paragraph and of the next one is not optimal.

For me, the critical issue you want to highlight here is the size of the pool of analogs candidate which has to be the largest as possible. Hence, MTW can increase this size (inflation). Another possibility relies on long archives of daily precipitation but requires estimates of sub-daily structures. > Here you can introduce the issue of the quality of the chronology of precipitation at a high resolution (e. 3hourly) time step.

Theses paragraphs were substantially edited.

As a perspective, you could also say that another (simpler) strategy would be to use a database relying on two different data sources :

-       MTW for the period with 3hourly data (30yrs * 8 equivalent data amount)

-       Classical fixed window approach for the period with only daily data (1900> 1980 = 80 years of additional daily data)

We added the following sentence in this regard: "Finally, another option could be to combine both the MTW and the classical approach and to look for analogue situations at a sub-daily time step from the 80's and at a daily time step for the antecedent years"

P13 – ln 32 : I do not really understand what you suggest as a perspective in the following paragraph.

"The precipitation data archives of high temporal resolution have increased over time. Other possible sources of such archives is the establishment of precipitation reanalysis at a regional scale in addition to the use of reanalysis-driven regional climate models or limited area models over a long period. Even though outputs from these models might be biased or not accurate enough, information regarding the timing of the precipitation events could be useful in disaggregating the station time series."

> I understand the potential interest of precipitation estimates from reanalysis-driven climate models over a long period. But, what do you mean with "establishment of precipitation reanalysis at a regional scale" and what is the difference / interest when compared to "reanalysis-driven regional climate models or limited area models over a long period". (why do you use the term "in addition") – what is typically the long period you have in mind ? Do you suggest to use these simulated precipitation data as a proxy to disaggregate daily precipitation observations ? Please Clarify.

We considered that this discussion goes too much into details for a perspective of secondary importance and we replaced the paragraph by "More advanced disaggregation methods might provide suitable sub-daily time series."

P14 – ln 4 > please clarify this paragraph.

I do not understand your statement : "this [selection] improvement has the potential for application to long meteorological archives."

Is the main idea to say that the MTW can be used to have a better diagnosis of the current / future weather situation even if no high resolution and high quality precipitation data are available ?

Is the "long archive" issue a key issue here ?

Could you explain why we do not need such quantitative values of precipitation for these analog dates ? (do you consider that you may have other observed values/events in some other historical database (flood events, other hydrometeorological proxies) that allow you to inform on the likely severity of the current weather situation to predict ?)

This paragraph has been substantially reduced and simplified.

You finally mention : "Finally, some other predictands might not need sub-daily total values but point observations such as hail or extreme wind gusts, which make them easier to use with the MTW." Do you suggest that the "better easiness" relies on the fact that each point observation can be attributed without any disaggregation issue to each of the different MTW window of a given day ? Please clarify or rephrase…

[revised manuscript text omitted]

---

## Author Response (AR4)

**Reply to editor**

Comments to the Author:

Thanks for the revisions. I now find your manuscript acceptable for publication, but would like you to correct a few minor issues as described below.

Best regards,

Jan Seibert

We would like to thank the editor for the positive feedback and for the time he dedicated to this manuscript.

P2L8, weather forecasting, flood forecasting, and hydropower production

Please split up the references so that it is clear which refers to what aspect

The references are now specific to the different uses.

P4L11: S1, 1 should be subscript (also P7L31 and other places)

This was corrected in the whole manuscript.

P4L15: The time of the day at which the predictors are selected

'for' instead of 'at'

This has been corrected.

P7L14 and P14L4: delete 'very' (we non-native speakers often use too many 'very', these two are examples of this)

This has been corrected.

P8: When adding the second level of analogy on moisture variables of the 2Z-2MI method (Table 2), the number of candidate situations for this level did not increase when using the original parameters because they were conditioned by the N1 previously selected analogues; however, their dates changed.

I am not sure I understand how the dates could change if 'they' were conditioned by previously selected analogues. Please clarify this sentence. (what exactly is 'they' referring to? It reads like 'parameters, but I think you mean 'analogues …. Sorry, confused ….)

[revised manuscript text omitted]